# Structural insights into the modulation of coronavirus spike tilting and infectivity by hinge glycans

David Chmielewski[1,8], Eric A. Wilson [2,8], Grigore Pintilie[3], Peng Zhao [4], Muyuan Chen[5], Michael F. Schmid [5], Graham Simmons[6,7], Lance Wells [4], Jing Jin [3,6,7] ✉, Abhishek Singharoy[2] ✉ & Wah Chiu [1,3,5] ✉

Coronavirus spike glycoproteins presented on the virion surface mediate receptor binding, and membrane fusion during virus entry and constitute the primary target for vaccine and drug development. How the structure dynamics of the full-length spikes incorporated in viral lipid envelope correlates with the virus infectivity remains poorly understood. Here we present structures and distributions of native spike conformations on vitrified human coronavirus NL63 (HCoV-NL63) virions without chemical fixation by cryogenic electron tomography (cryoET) and subtomogram averaging, along with site-specific glycan composition and occupancy determined by mass spectrometry. The higher oligomannose glycan shield on HCoV-NL63 spikes than on SARS-CoV-2 spikes correlates with stronger immune evasion of HCoV-NL63. Incorporation of cryoET-derived native spike conformations into all-atom molecular dynamic simulations elucidate the conformational landscape of the glycosylated, full-length spike that reveals a role of hinge glycans in modulating spike bending. We show that glycosylation at N1242 at the upper portion of the stalk is responsible for the extensive orientational freedom of the spike crown. Subsequent infectivity assays implicated involvement of N1242-glyan in virus entry. Our results suggest a potential therapeutic target site for HCoV-NL63.

The coronavirus surface is studded with spikes that initiate infection by binding to specific cellular receptor(s) leading to membrane fusion and viral entry. Three copies of spike glycoprotein (S) form a mushroom-shaped spike. The S-receptor interaction determines cellular and species tropism and influences pathogenicity. Importantly, mutations in receptor binding motifs (RBMs) are responsible for cross-species transmission of coronaviruses. Current mRNA vaccines for SARS-CoV-2 encode stabilized soluble S trimer in the prefusion conformation to elicit neutralizing antibody responses to protect host from virus infection[1,2]. A myriad of antibody- and protein-therapeutics have been designed to block S binding to the human ACE2 (huACE2) receptor[3–8].

Each trimeric spike has a globular crown and a thin stalk anchored to the viral envelope. Structures of coronavirus crowns in the prefusion state, including members of alpha-, beta-, gamma- and delta-coronaviruses, have been extensively characterized following

[1]Biophysics Graduate Program, Stanford University, Stanford, CA 94305, USA. [2]School of Molecular Sciences, Biodesign Institute, Arizona State University, Tempe, AZ, USA. [3]Department of Bioengineering, and of Microbiology and Immunology, Stanford University, Stanford, CA 94305, USA. [4]Complex Carbohydrate Research Center, University of Georgia, Athens, GA 30602, USA. [5]Division of CryoEM and Bioimaging, SSRL, SLAC National Accelerator Laboratory, Stanford University, Menlo Park, CA 94025, USA. [6]Vitalant Research Institute, San Francisco, CA 94118, USA. [7]Department of Laboratory Medicine, University of California, San Francisco, San Francisco, CA 94143, USA. [8]These authors contributed equally: David Chmielewski, Eric A. Wilson. ✉e-mail: jjin@vitalant.org; asinghar@asu.edu; wahc@stanford.edu

expression, purification and cryogenic electron microscopy (cryoEM) analysis of soluble constructs of S trimer without the stalk and the transmembrane region[9]. Each S protomer consists of a N-terminal S1 subunit and a C-terminal S2 subunit. For HCoV-NL63, the S1 subunit is made up of 5 smaller subdomains: 0, A, B, C and D, with domain B acting as the receptor binding domain (RBD) for ACE2[10,11]. SARS-CoV[12], SARS-CoV-2[13,14] and HCoV-NL63[15] all infect host cells via binding to the receptor huACE2 by the RBD in the S1 C-terminal domain (S1-CTD)[9]. The RBD in S1 sits at the top of the prefusion crown, with S1 positioned above S2, the virus fusion machinery. After receptor binding and protease cleavage at the S2′ site[9], S1 dissociates from S2 and S2 transitions from prefusion to postfusion states to drive fusion of viral membrane and cellular membrane. In contrast to multiple conformations of RBDs in prefusion spikes of SARS-CoV, SARS-CoV-2 and MERS-CoV[16–20], the prefusion spikes of many coronaviruses in other genera display a single closed conformation with RBD unable to bind its receptor[10,11,21–23]. The mechanism of activating the RBD binding is not known for HCoV-NL63. The spike crown is highly N-glycosylated, with glycans likely shielding S epitopes from the host immune response. Due to its extreme flexibility and predicted intrinsically disordered regions, there are currently no high-resolution structures of the stalk and its attached glycans. Recent structural studies of chemically fixed SARS-CoV-2 virions revealed the stalk accommodates bending motions of the upper regions of the crown and a model of flexible stalk hinges was proposed[19,24]. It is not understood if chemical fixatives used to inactivate SARS-CoV-2 would alter the observed S conformational dynamics. The conformational landscape of full-length S in situ without the influence of chemical fixation, a structural mechanism of S conformational variability that produces the observed conformational states, and the functional implications of S flexibility for infection and/or immune escape remain uncertain[24,25].

A more complete understanding of S in initiating cell entry and interacting with the immune system requires characterization of structure and function of the full-length S on the virus surface. Here, we addressed such questions by imaging infectious human coronavirus NL63 (HCoV-NL63) particles without chemical fixative using cryogenic electron tomography (cryoET) to determine the conformational landscape of full-length S trimers on the virion surface through subtomogram averaging and 3D visualization. We used mass spectrometry to characterize site-specific microheterogeneity of glycosylation at every predicted N-linked glycosylation site on HCoV-NL63 S and modeled them in our cryoEM and cryoET maps. All-atom molecular dynamics (MD) simulation of the full-length S extending beyond the viral membrane was carried out to estimate the probability distribution of S conformational dynamics. The simulation results not only agreed with the structural landscape of S from cryoET but also guided mutagenesis experiments of glycan-associated amino acids around the stalk-crown interface. This demonstrated the critical roles of the hinge glycans in human coronavirus NL63 infection.

## Results

### Structural landscape of HCoV-NL63 spikes presented on virion surface

We previously reported a 3.4 Å single-particle cryoEM structure of HCoV-NL63 S crown by imaging vitrified virions without chemical fixation, purified from the supernatant of virus-infected MA104 cells through iodixanol density gradient ultracentrifugation[10]. In the single-particle cryoEM map, the stalk region was not resolved beyond residue 1225 towards the C-terminus. In the current study we used cryoET and subtomogram averaging to determine the structures of the full-length S ectodomain on virions.

Following vitrification of chemically unfixed HCoV-NL63 virus particles, a cryoET dataset of 176 tomograms was collected from which 18,356S trimer subvolumes were extracted automatically (Fig. 1A; see "Methods"). Using manual picking and inspection, we randomly

selected 154 virions and counted the number of spikes per virion to be ~20+/−13. Additional visual inspection of the tomograms clearly revealed variable conformations of spikes (Supplementary Movie 1). Subtomogram averaging analysis of these spikes with imposed C3 symmetry produced an average at 6.9 Å resolution for the crown and lower resolution ~11Å for the upper part of the stalk (Fig. 1B; Supplementary Figs. 1–2). Secondary structure elements like alpha helices are clearly resolved in the crown density (Fig. 1B). These secondary structure elements are closely matched to those in our previously published model of the crown in the pre-fusion conformation (PDB:7KIP) that missed the stalk region entirely[10]. The density below the crown continues until sharply reduced resolvability at a knob-like density 48 Å below the endpoint of our atomic S model determined in the previous single particle cryoEM map (EMD-22889). This density resolvability deterioration is presumably due to extreme orientational flexibility of the spike crown relative to the stalk region just below as shown in Fig. 1C, where the subtomogram averages were mapped back to the raw spike density in the tomogram.

To further assess any conformational variability within the S crown, we applied focused classification on both the RBD region and domain 0 region of the three S protomers within each trimeric spike. This analysis confirmed that all three S protomers in a spike are in the pre-fusion, "closed" conformation with all RBDs lying down and no "upward" conformation of domain 0 was detected (Supplementary Fig. 2). Such conformational stability of the crown is consistent with previously published structures of purified soluble spikes of alpha-coronaviruses[11,21,26,27].

Separately, processing of the stalk and viral membrane region using focused classification yielded an average map with a resolution of 11.5 Å, with imposed C3 symmetry (see "Methods", Fig. 1D). The linear density protrudes ~100 Å perpendicular to the membrane before a sharp cut-off. The small bulging density on the lower stalk near the membrane is consistent with predicted N-linked glycans at N1277. On the virion surface, the average distance between the nearest neighboring spikes is ~34 nm (Fig. 1E) compared to ~15 nm average nearest distance between prefusion spikes on SARS-CoV-2 virion[19].

Spike crowns were often visualized in highly tilted conformations relative to the stalk that connects to the viral membrane (Fig. 1F, G, Supplementary Fig. 3A, Supplementary Movie 1). The S on purified HCoV-NL63 virion displays a single fuzzy band of an apparent 250 kD after separation through SDS-PAGE (Supplementary Fig. 3B), which is consistent with the lack of multibasic furin cleavage site at the S1/S2 boundary of HCoV-NL63 S. The structural flexibility of spike prevents resolving the full-length S ectodomain structure in subtomogram averages, with orientations determined by either the crown or stalk region alone (Fig. 1B, D). Instead, by using the orientations of crown and stalk regions for each spike, it was possible to piece together a stalk+crown conformation for each spike. To compute subtomogram averages of full-length spike conformations we needed to group them together by two different criteria: tilting angle of the crown relative to stalk (tilted), and azimuthal direction, calculated by determining the direction of the crown tilt relative to the central axis of the stalk. Subtomogram averages of the full-length spike ectodomain could be classified into eight distinct tilted conformations (Fig. 1G) and six conformations showing different azimuthal direction of tilt (Supplementary fig. 3C). Spike crowns could be tilted up to 80° relative to the stalk, with a median tilt of $41.6 \pm 15°$ (Fig. 1F). The calculated azimuthal direction of tilts ranges from 0° to 120° with C3 symmetry imposed (equivalent to 360° without symmetry), and the distribution shows roughly equal probability of spikes assuming all angles (Supplementary Fig. 3D). The averages reveal the significant conformational variability of the trimeric spike through tilting and rotation that are mediated primarily by a large hinge localized to an upper region of the stalk ~100 Å away from the viral envelope (Fig. 1F). Since the spike crowns showed a clear preference for tilting, but no preference for the

azimuthal direction of that crown tilt relative to the stalk, we focused our later analysis on the tilting motions mediated by the stalk hinge.

## HCoV-NL63 spikes are covered by a dense, oligomannose dominant glycan shield

Coronavirus spikes are heavily glycosylated, and glycosylation is known to play critical roles in virus infection and immunity development[28]. Recently, S1 glycans were reported to play a role in

regulating the conformational transitions of SARS-CoV-2 RBD[29,30] or the PEDV domain 0[31]. To understand if any of the observed conformational variance of the native HCoV-NL63 spike on virus surface was regulated by such glycans, we next sought to identify glycan compositions at each N-glycosylation site and generate the fully glycosylated spike model for all-atom molecular simulations of the S trimer dynamics outside the viral membrane. We first applied LC-MS/MS analysis to glycopeptides generated by digesting S proteins

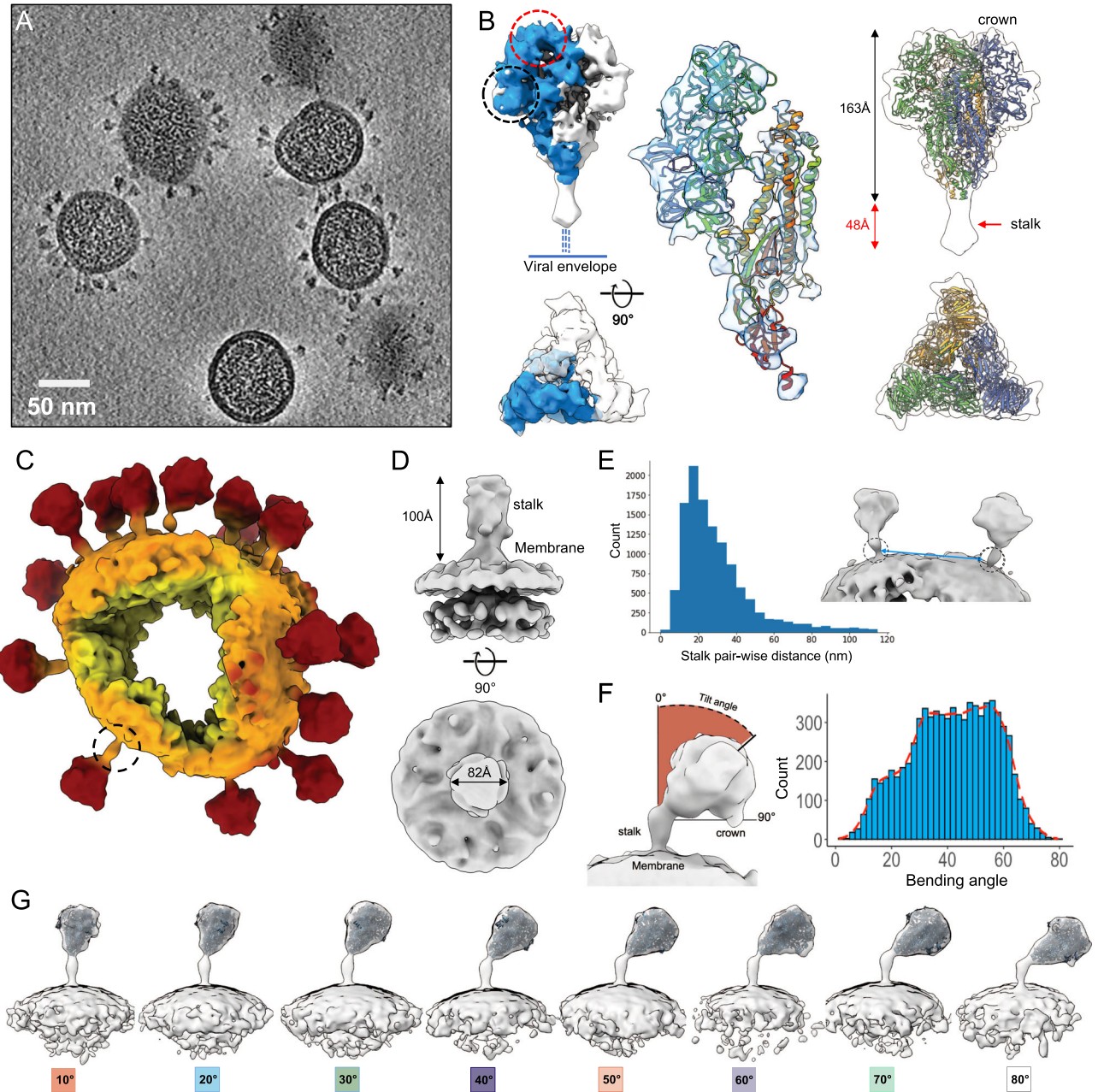

**Fig. 1 | CryoET of HCoV-NL63 intact virions. A** Tomogram slice of HCoV-NL63 virions embedded in vitreous ice. **B** Subtomogram average of prefusion spike crown at 6.9 Å resolution. Density map is displayed from the side view (top) and top view (bottom), with the density of one S protomer colored (blue) and RBD (red circle) and domain 0 (black circle) regions marked. Zoom-in view of one S protomer with atomic model (PDB: 7KIP, rainbow) fitted into the density. Spike crown sub-tomogram average, with fit atomic crown model, shows additional density in the upper stalk region (red arrow) previously unresolved in the single particle cryoEM map (EMD-22889). **C** Colored density of a single virion with lipid envelope (orange) and spikes (red) in actual orientations. One spike stalk is circled in a dotted line.

**D** Subtomogram average of lower region of spike at 11.5 Å resolution, with the lower portion of the stalk and viral envelope. **E** Distribution of measured nearest-neighbor distances between spike stalks on virions. Two neighboring spike stalks are circled in dotted lines and the distance between them is labeled in a blue line. **F** Distributions of bending angle of crown relative to stalk for each spike, determined from the refined subvolume orientations in discrete (blue) and continuous representation (red). **G** Structures of full-length spikes grouped according to bending orientation of the crown relative to the stalk region. Atomic model of S crown (PDB: 7KIP) fits into density for visualization. Source data are provided as a Source Data file.

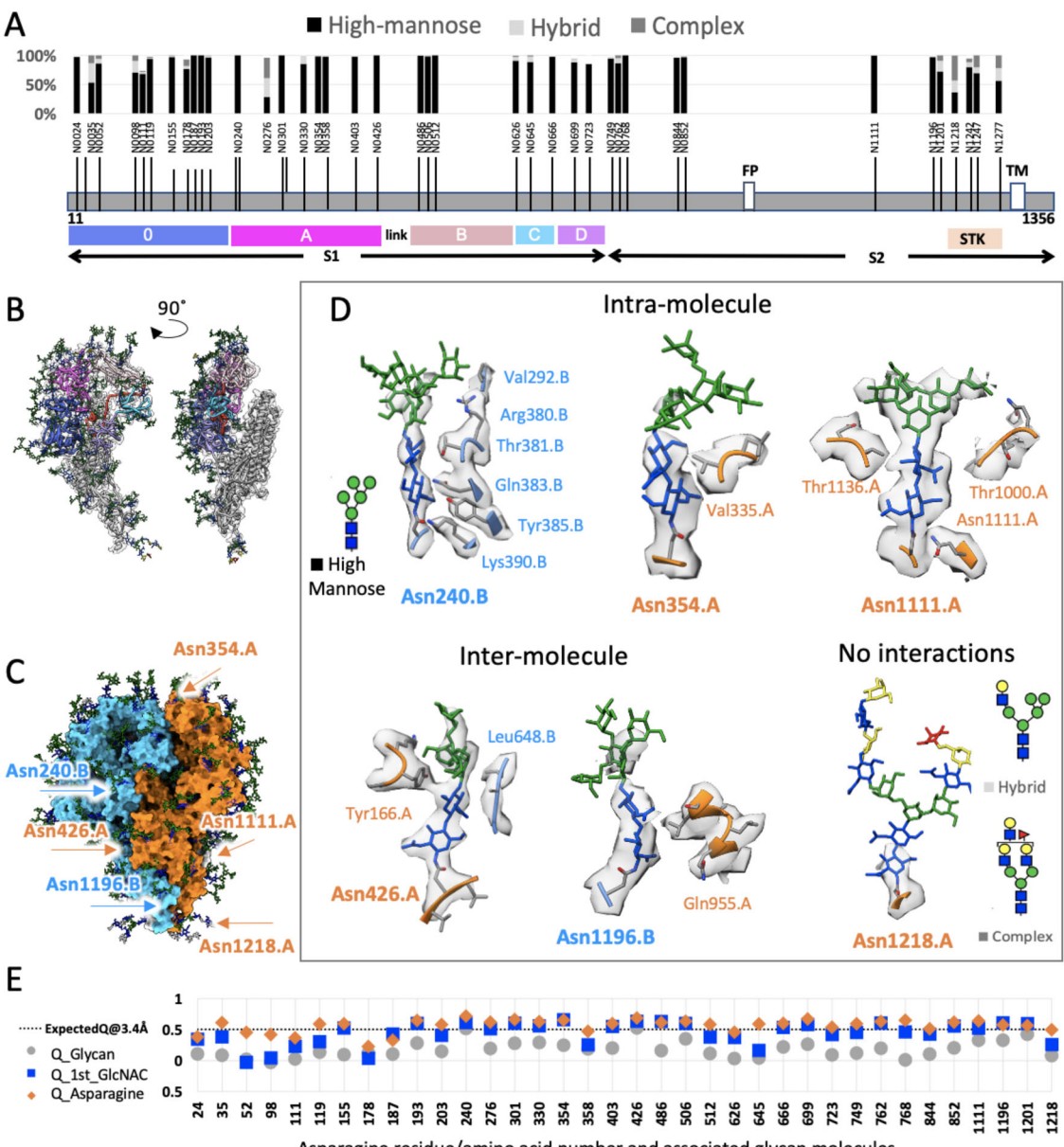

**Fig. 2 | Structure of the fully glycosylated HCoV‑NL63 spike. A** Bar plot showing occupancies of glycan types at Asn residues of 39 N‑glycosylation sequons in the HCoV‑NL63 spike protomer determined by mass spectrometry. **B** 5 Domains (0, A, B, C, D) in S1 are colored within a single protomer, shown from two angles. **C** Three protomers in the spike are colored with blue, orange, and white surfaces. **D** Molecular interactions between glycans and nearby amino acid residues for representative N‑linked glycans associated with either protomer P1 (orange) and/or P2 (blue) of the trimer (designated by the residue numbers) are shown in the 6 panels. The schematics of three types of glycans: high mannose, hybrid and complex are shown. **E** Q‑scores calculated for the 36 Asn residues and associated glycans in the crown; three points are shown: orange diamond represents Q‑score for Asn residue, blue square represents the Q‑score of the first monosaccharide unit attached to Asn (GlcNAC), and gray circle represents average of Q‑scores for all monosaccharides attached at that residue position. Source data are provided as a Source Data file.

extracted from purified HCoV‑NL63 virions with different combinations of proteases to maximize glycopeptide coverage (Supplementary Data 1). We were able to determine the microheterogeneity in each of the 39 canonical N‑linked glycosylation sites on HCoV‑NL63 S. Based on the assignment and spectral counts for each topology, we could determine site‑specific occupancy and percent of the N‑linked glycan types (high‑mannose, hybrid and complex) present at each site (Fig. 2A).

By building an atomic model of the most abundant glycan topology into the density at each site of our previously‑reported 3.4 Å single particle cryoEM map of the crown, we found some of the high‑mannose glycan chain densities were well resolved as demonstrated by

visual and Q‑score quantification[32] (Fig. 2B–D, Supplementary Fig. 4, Supplementary Movie 2), while others with complex glycan were only resolved in the cryoEM map density proximal to the amino acid side chains[10]. Of the 39 N‑glycosylation sites, 36 are on the crown portion resolved in our cryoEM map (EMDB‑22889), and display densities that cover at least the first N‑acetylglucosamine moiety (Figs. 2B, 1C & Supplementary Fig. 4). The remaining 3 glycans are in the unresolved stalk region. Of the 36 on the crown, 29 are predominantly high‑mannose glycans (>80%) (Figs. 2A, and 3A, B), consistent with reported N‑linked glycan profile on HCoV‑NL63 S ectodomain produced in *Drosophila* S2 cells[11]. The predominant high‑mannose glycan structures at the 29 sites are $Man_{7-9}GlcNAc_2$, suggesting that steric

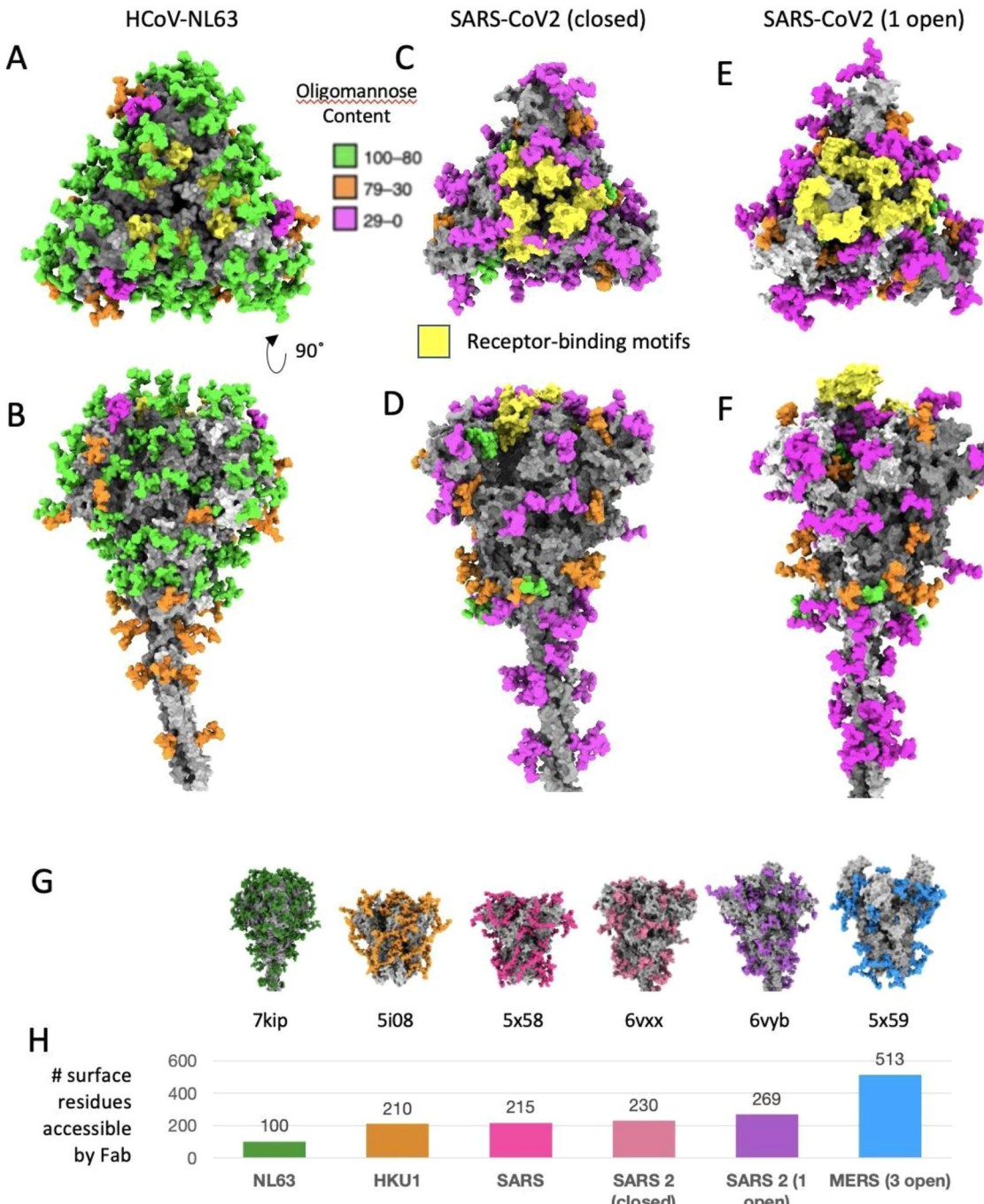

**Fig. 3 | Abundance, distribution of high mannose glycans and glycan shield densities on glycoproteins of human coronaviruses. A**, **B** Top and side views of HCoV-NL63 trimeric spike; protomers are shown with gray surface, glycans are shown with green, orange, or purple colors depending on percentage of oligomannose content. The receptor-binding motifs are colored in yellow. **C**, **D** Same color coding for SARS-CoV 2 spike protein closed conformation[30], and **E**, **F** conformation with one protomer in the open conformation[30]. **G** Models of different coronavirus spikes and associated PDB IDs, with full glycans modeled (deposited models contain only partial glycan models). **H** Number of surface residues accessible to Fab - lower numbers suggest more immune evasiveness of the virus. Source data are provided as a Source Data file.

constraints between the glycans and their surrounding spike protein moiety prevent the access of α-mannosidase I in the Golgi apparatus to cleave off the distal mannose at these sites. In marked contrast, high-mannose glycans are present on only 2 of 22 (SARS-CoV), 4 of 22 (SARS-CoV-2), 7 of 23 (MERS-CoV), and 3 of 26 (HCoV-KHU1) N-glycosylation sites on other coronavirus spikes[33,34]. This is consistent with the stable and compact conformation of HCoV-NL63 spike crown in contrast to flexible open-close conformational transitions in spikes of other coronaviruses[10,16,20].

To understand why some of the glycan densities are well resolved while others are less resolved in the cryoEM map, we scrutinized the chemical environments of the glycans. Highly resolved glycans with higher Q-scores are found to interact with surrounding amino acid residues in the same and/or neighboring S protomer. Those with lower Q-scores are found to have no interaction with the S protein beyond the Asn attachment site. This suggests that stabilization of those glycans occurs via sugar-protein interactions, as exemplified in glycans attached to Asn240, Asn354, Asn426, Asn1111, Asn1196, via

intermolecular or intramolecular interactions (Fig. 2C, D). The N240-linked glycan has the highest Q-score among the all 39 N-linked glycans (Fig. 2E) and is highly conserved across alpha- and delta-coronaviruses (Supplementary Fig. 5A). It inserts between the top galectin-like domain and the bottom three antiparallel β-sheets of the S1 domain A (Supplementary Fig. 5B). This sandwiched glycan may help stabilize the surrounding protein domains, as well as explaining its stable presence in this location.

### Integrative modeling of HCoV-NL63 full spike ectodomain

Using single-particle cryoEM and cryoET we have produced density maps of the HCoV-NL63 spike at different resolutions and in multiple conformations. To precisely locate the hinge in the spike structure and determine the molecular origin of the tilting observed in the bent spike conformations, a full-length model of the spike ectodomain is constructed. The near-atomic resolution single particle cryoEM density map of the crown[10] allows atomic model building of the protein and associated glycans from residues 1 to 1224, which includes the first nine residues (1216–1224) of the stalk (Fig. 2B). However, the low resolution (11.5 Å) of the remaining stalk density in the cryoET map (Fig. 1D) of the crown does not allow deterministic atomic model building. To generate a full-length model of the spike ectodomain (residues 1–1297), we establish a probable model of the full-length stalk from residues 1216–1297.

Detailed in the Methods section, a predicted model of the stalk monomer (residues 1216–1297) was generated using I-TASSER[35] and cross validated with AlphaFold (Supplementary Fig. 6). Three such monomeric stalk models were grafted onto the full atomic crown model by aligning their short alpha helical region (residues 1216–1224) with the higher resolution single particle model (PDB: 7KIP). The entire trimeric spike assembly was energetically-optimized using Molecular Dynamics Flexible Fitting (MDFF) within the recent cryofold setup[36]. Following further optimization of the helix contacts using CC Builder, high-mannose glycans were assigned to N1242 and N1247 on the stalk closer to the crown, while a complex glycan was assigned to N1277 further away from the crown by using the CHARMM-GUI interface[37], based on our mass spectrometry assignments and a previous report[11].

The glycosylated, full-length model of the spike ectodomain was employed to generate seven individual models at tilt or bending angles 10°, 20°…70° corresponding to the spike subtomogram class averages, by using 30 ns of MDFF for each model (Fig. 1G). The final fit of these models into the density maps revealed that a disordered loop region (approximately residues 1226–1245) separating the two coiled-coil motifs in the stalk was consistently placed near the hinge region (Supplementary Fig. 6).

### Estimation of glycan shield in coronaviruses

High-mannose clusters predominantly cover the crown of the HCoV-NL63 spike (Fig. 3A, B). In HCoV-NL63 S the receptor binding motifs (RBMs) are predicted neutralizing epitopes[15] and are hidden by the interface between A and B domains of S1. The RBM motifs are also shielded by the top N-linked glycans at N486, N506 and N512 on RBD and N358 at the interface between domain A and domain B (Supplementary Fig. 7A). This is reminiscent of the high-mannose clusters on viral glycoproteins of classic immune evasive viruses, such as HIV-1 Env[38] and LASV GPC[39]. Additionally, a cluster of N-linked glycans at N844, N852 and N749 on a neighboring S protomer protects the activation loop upstream of the S2′ cleavage site (Supplementary Fig. 7B), suggesting it forms a glycan gate to tightly regulate the cleavage at S2′ that activates the fusion peptide.

In contrast to the HCoV-NL63, the SARS-CoV-2 spike has fewer high-mannose glycans and has its RBD exposed to the surface in either closed or open conformations (Fig. 3C–F). The dense glycan shield on the surface of HCoV-NL63 spike suggests efficient epitope masking and immune evasiveness from antibody recognition. To quantify it, we calculated solvent-accessible surface area (SASA) on spikes of a panel of different coronaviruses (Fig. 3G, H, Supplementary Fig. 8), with a spherical probe radius of 20 Å, which roughly corresponds to the complementarity-determining regions (CDRs) of a Fab fragment of an antibody. The number of accessible residues depends on the number of glycans and their distribution on the surface of each virus spike. For example, for HCoV-NL63, the number of accessible residues without glycans is 1431, whereas with glycans it is only 100 residues. The number of residues accessible by Fab estimated for different coronaviruses are plotted in Fig. 3H. HCoV-NL63 spike has fewer residues accessible by Fab than other coronaviruses spikes, suggesting HCoV-NL63 is more immune evasive than other coronaviruses.

### Simulation of HCoV-NL63 Spike reveals the correlation between hinge bending and epitope shielding

With seven different full-length models of the glycosylated spike ectodomain resolved at tilt angles of 10°, 20°…70° (Figs. 1G and 4A (inset)), we next investigated the molecular origins of the observed bending motions. Conventional MD simulations were performed starting from the integrative models constructed at the seven bending angles to sample the energetics of the entire bending landscape. Each of the fitted models were individually subjected to 100 ns of explicit followed by 350 ns implicit solvent simulations, resulting in 3.15 μs of total sampling time across the seven independent simulations (See "Methods") to observe the bending dynamics of the HCoV-NL63 stalk. In the simulations, spike bending was defined as the angle between the principal axis of the crown and that of the long stalk coiled-coil region (see methods and Supplementary Fig. 9). The bending probability derived from these simulations of the full-length, glycosylated spike matched the overall experimental distribution of spike bending on intact virions as determined by subtomogram analysis (Fig. 4A and Supplementary Fig. 10A). Both distributions displayed peaks at the same high bending angle (~56°), also exhibiting a similar range. And no correlation between azimuthal angles and bending angles of the spike was detected (Supplementary Fig. 11). The agreement between experimental and in silico simulation of spike bending ensemble dynamics suggested our MD simulation system was aptly capturing the energy landscape of the spike in situ. From our simulation results, we determined that most of the bending was localized to the disordered region (residues 1226–1245) separating the two coiled-coil motifs of the stalk (Supplementary Fig. 10B, C). Two N-linked glycans (attached to N1242 and N1247) reside near this disordered hinge region (Fig. 4B).

Our glycomics analysis revealed a cluster of processed N-glycans (hybrid and complex glycans) at the C-terminal base of the crown (N1201 and N1218) and the stalk region (N1247 and N1277) (Figs. 2A–C and 3B), suggesting a structural flexibility in these regions allowing access of glycan processing enzymes, in contrast to the stable compact crown of HCoV-NL63 spike. This is consistent with the large range of spike bending angles (Fig. 1F, G) and low-resolution features of the spike stalk in our subtomogram averages (Fig. 1D). In our stalk model, N-linked glycans at N1242 and N1247 sit between the upper short coiled-coil and lower long coiled-coil and likely contribute to the blob-like density at the bending hinges in spike subtomogram averages (Fig. 1F, G). The upper short coiled-coil region in HCoV-NL63 spike is equivalent to the stem-helix in SARS-CoV-2 spike (Fig. 4B) that is the epitope targeted by several recently identified broad neutralizing antibodies against betacoronaviruses[40–47]. Thus, this coiled-coil region in HCoV-NL63 spike could be a critical neutralizing epitope. Akin to SARS-CoV-2, where N1158-linked glycans shield the stem-helix epitope, the N1242- and N1247-linked glycans potentially serve a similar purpose in NL63.

To probe the epitope-shielding potential of the glycans adjacent to the stalk hinge (N1242 and N1247), we performed maximum accessible surface area (ASA) analysis of the simulated models. Illustrated in Fig. 4C, in the bent conformation the hinge glycans shield

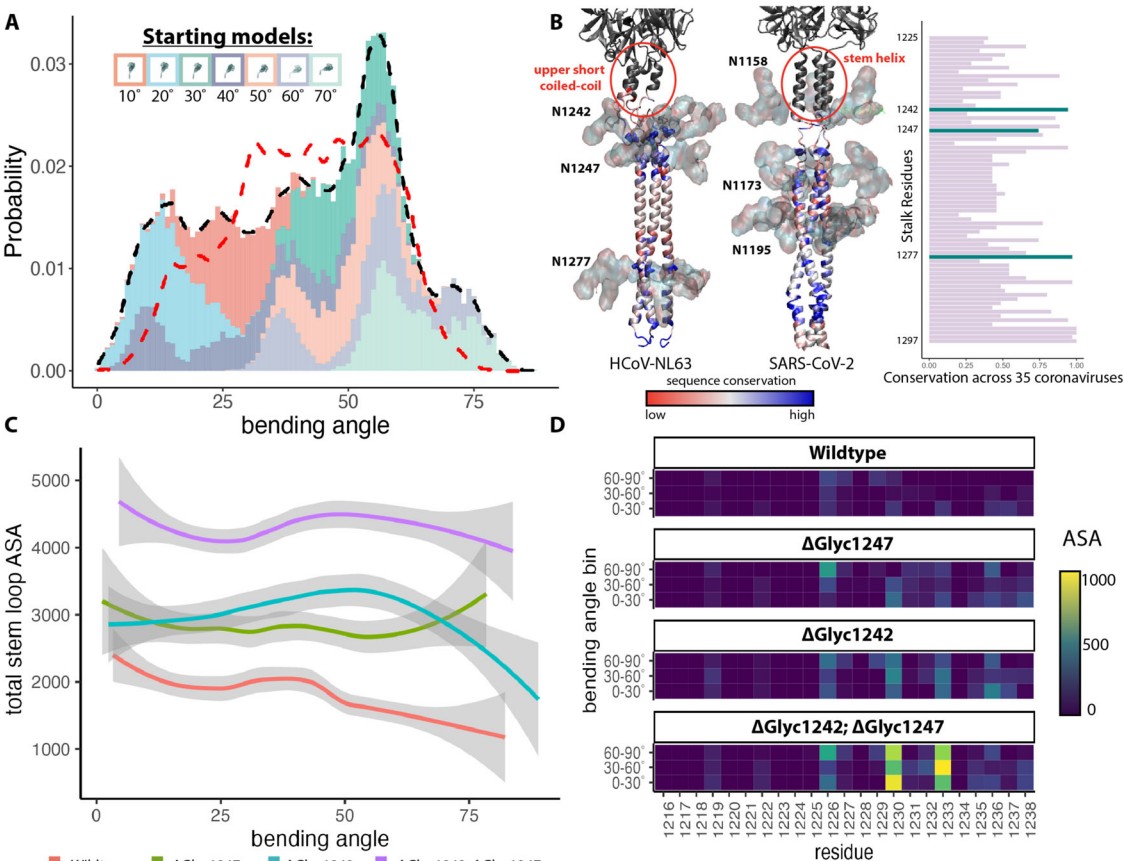

**Fig. 4 | Molecular simulation of glycan shielding of stalk epitopes. A** The simulated bending profile of the HCoV-NL63 spike. Different colors represent different starting models, as indicated by the legend (they are also displayed separately in Supplementary Fig. 8A). The black trace shows the overall bending angle distribution calculated from the MD simulations, while the red one is identical to the distribution observed in the tomograms in Fig. 1F. **B** Conserved glycosylation sites in coronavirus stalks. Left: The stalk regions of HCoV-NL63 and SARS-CoV-2 are colored by amino acid sequence conservation with blue regions indicating highly conserved residues and red regions indicating highly variable regions. Right: A multiple sequence alignment of S genes from 35 coronaviruses across all genera

indicates the conservation (x-axis) by stalk residue (y-axis), with the highlighted bars (green) indicating the three highly conserved glycosylation sites. **C** The accessible surface area (ASA) of the stem helix region (1216–1238) of the HCoV-NL63 stalk to antibody CDR-sized molecules was measured using a probe radius of 2 nm. The total ASA of the stem helix region as a function of bending angle is shown for the wild-type spike model as well as simulations of models with various hinge glycans removed. The predicted values ± 1.9 x standard errors are presented. **D** Glycan simulations were binned into 3 groups based on bending angle: 0–30°, 30–60°, and 60–90°. The median ASA of each residue across each bin is presented as the heatmap. Source data are provided as a Source Data file.

the predicted neutralizing epitope from access by any protein of the size of complementarity-determining region (CDR) of an antibody. Interestingly, there was minimal epitope accessibility, suggesting high shielding, near the most probable bending of 56˚, which also corresponds to the hinge glycan-protein interaction energy global minima (Supplementary Fig. 12A, B top plots).

The varied epitope shielding by hinge glycans at different spike bending angles (Fig. 4C, D) prompted us to further analyze steric interactions between stalk glycans and surrounding protein regions. Supplementary Fig. 12C presents ensembles of 1834 atomic conformations of the hinge glycans (N1242,1247) at bends of the stalk between tilt angles 0°–30°, 30°–60° and 60°–90°. Visualization of these ensembles reveal that the conformational space of the two glycans overlap when the tilt ranges between 0°–30°. At higher tilt values, the hinge bending dynamics is associated with separation of the two glycans conformations into distinct clusters. Between 30° and 60°, while the N1242 glycan is near the base of the crown, the N1247 interacts with the stalk. A heatmap of contact between the hinge glycans and the upper stalk region depicts stronger interactions at angles 30°–60° relative to the bends between 0°–30° and 60°–90° (Supplementary Fig. 12C). Taken together, the most favorable protein-glycan interactions (Supplementary Fig. 12A) computed at the tilt range of

30°–60° is consistent with the stabilization of the spike at ~56° bent conformation, as well as the minimum ASA of the epitope that is also observed at this angular range (Fig. 4C).

**Deletion of hinge glycans produces deviations in bending profile**
Given our structural and chemical observations, we hypothesized that glycans near the hinge region interacting with nearby protein surfaces on the S crown could influence overall bending and structural conformation of the spikes. The difference in the contribution of N1242- and N1247-linked glycans to glycan-protein interactions at different spike bending angles (Supplementary Fig. 12A, B) suggests they may affect spike conformations differently. So, we next performed simulations exploring the role of N1242- and N1247-linked glycans in S bending (Fig. 5A, B, Supplementary Fig. 13 and Supplementary Movie 3).

A cumulative 2.1 µs implicit MD simulation was repeated with five different glycan removal strategies (N1242-glycan and N1247-glycan single deletion or double deletion, removal of all spike glycans, and removal of all glycans except the two hinge glycans) to probe the influence of glycans on conformational dynamics of the ectodomain of full-length spike. The removal of both the hinge glycans, N1242 and N1247, from the S trimer resulted in the simulations exploring

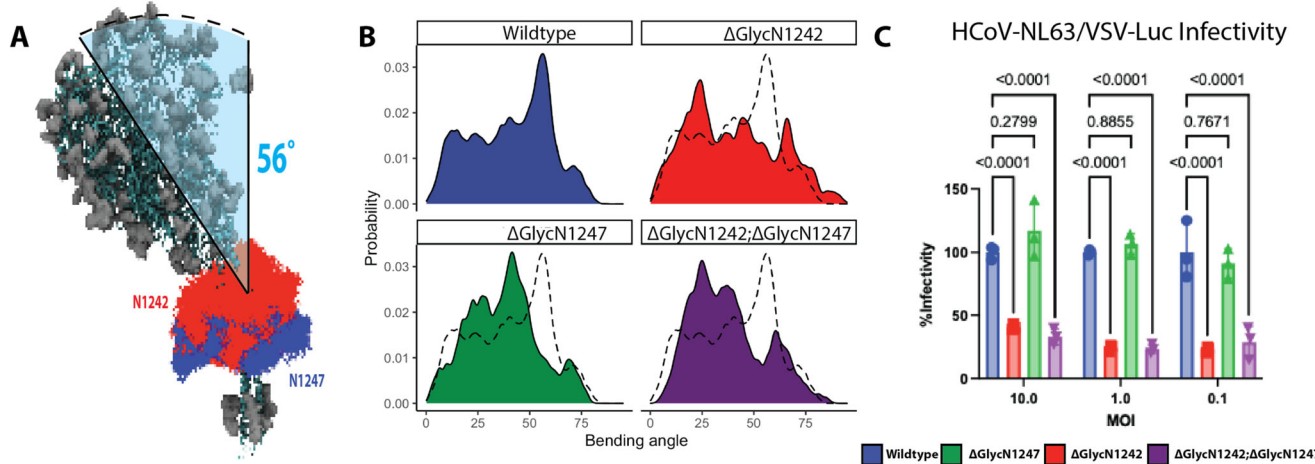

**Fig. 5 | Highly conserved hinge glycans modulate HCoV-NL63 spike bending.**
**A** N1242 (red) and N1247 (blue) glycans were extracted from WT spike MD trajectories and overlaid on a representative tilted structure. **B** The bending angle distributions for 5 different glycan modifications. The black dotted line indicates the WT bending angle distribution. **C** Functional infectivity assays for various stalk glycan modifications. Single or double deletion of N-linked glycan at the residues 1242 and 1247 were generated by mutating Asn to Asp in HCoV-NL63 S gene. HEK293-ACE2-TMPRSS2 cells were infected with wild type and mutant spike pseudotyped VSV-Luc reporter viruses at indicated MOI in triplicates. Viral infectivities were quantified by intracellular luciferase activities at 16 h post infection and normalized to the mean infectivity of wild type triplicates. Each bar represents the mean with SD of the triplicate samples. Data from one representative experiment of three independent ones were shown. Data were analyzed by two-way ANOVA analysis with correction for multiple comparisons by Turkey test. Source data are provided as a Source Data file.

shallower bending angles (peak at 25°) and exhibiting straighter conformations than the wild type (WT) stalk (Fig. 5B). Simulations with the N1242 glycan removed alone bent at comparable angles (Δgly1242 peaks at 20°–25°), which are smaller relative to those with the N1247 glycan removed alone (Δgly1247 peaks at 40°–45°). The statistical significance of this glycan-induced shift in bending angle populations relative to that of the WT is manifested by a miniscule $p$-value of $<2.2 \times 10^{-16}$ derived from an one-sided Mann–Whitney $U$ test[48]. Hence, we infer that the dynamics of N1242-linked glycan is most strongly coupled with that of the trimeric spike. Interestingly, simulation of the S trimer with all glycans removed offered the most straight conformation (peak at 20°), which also approaches the bending of the Δgly1242 closer than the Δgly1247 (Supplementary Fig. 13). Consistently, removal of the hinge glycans from the stalk markedly increased access to the stem helix epitope, suggesting an immune evasion role for the hinge glycans. Stalk bending, on the other hand, is facilitated by glycosylation, and further decreases access to this critical epitope (Fig. 4C).

**Highly conserved hinge glycan promotes bending and coronavirus infectivity**

To understand the mechanism of how N1242-linked glycan influences HCoV-NL63 spike bending, we further analyzed steric interactions between the hinge glycans and the unstructured region in the upper stalk protein regions and determined the interaction energy using the NAMD energy score function[49] (Supplementary Fig. 12). The interaction hotspots (indicated in red) show remarkable similarity between the WT spike and the Δgly1247 construct, suggesting that N1247-linked glycan contributes minimally to protein-glycan contacts. The interaction energy trends indicate that the most probable bending angles are more similar between the WT and Δgly1247 than between WT and Δgly1242 (Supplementary Fig. 12A, B). In stark contrast, removal of N1242-linked glycan decreased the protein-glycan contacts by 2-3-fold and pushed the most energetically favorable bending to lower tilt angles (Supplementary Fig. 12B). Probing deeper into the individual interactions of glycans attached to N1242 and N1247, we determined the energetic contributions of each hinge glycan as a function of bending angle (Supplementary Fig. 12A). We found that both glycans

had similar interaction profiles for moderate bending angles (20°–60°). However, bends >60°, while stabilizing the N1247 glycan-protein interactions, concomitantly weaken the N1242 glycan contacts with the upper stalk region. Altogether, the individual protein-hinge glycan energy landscapes essentially reinforce an energy minimum ~56°, matching the most populous angle in overall bending simulation. Energy barriers in individual glycan energies on either side of the minimum reinforces the most probable bending angle.

A multiple sequence alignment of S from 35 different coronaviruses spanning alpha-, beta-, gamma- and delta-coronavirus genera revealed that the N-linked glycosylation sequeons at the hinge (N1242 and N1247) are highly conserved (Fig. 4B and Supplementary Fig. 10D). N1242 and N1247 are 89% and 74% conserved, and residues at 1244 and 1249 are 91% and 69% conserved as Serine or Threonine. Thus, the N-linked glycosylation sequeon at residue N1242 is 89% conserved compared to N1247 glycosylation (57%). The high conservation of the hinge glycans across all four coronavirus genera suggests their potential functional relevance, and we tested the importance of the N-linked glycan at the spike hinge in viral infection by a pseudotype virus infectivity assay (Fig. 5C). We mutated Asn to Asp at the position N1242 or N1247 alone or both to remove N-linked glycans and generated wt and mutant spike pseudotyped VSV-Luc reporter viruses. HEK293 cells stably expressing huACE2 and TMPRSS2 were infected with HCoV-NL63 S pseudoviruses at MOI 0.1, 1.0 and 10. Viral infectivity was quantitated by measuring luciferase activity in cell lysates at 16 h post infection and normalized to the infectivity of wt S pseudotyped VSV-Luc virus. Removal of N1242 glycan alone or both N1242 and N1247 glycans reduced HCoV-NL63 S mediated infection by ~70% at three different MOIs, in contrast to the negligible effect of removing N1247 glycan alone. In conclusion, we uncovered a role of the conserved N-linked glycan at the hinge position N1242 in both the spike structure and infectivity of HCoV-NL63.

## Discussion

In this work, we directly studied spike structures on the purified intact HCoV-NL63 virions using an integrative approach including cryoEM/ET without chemical fixation, MS glycomics and MD simulations. We only visualized HCoV-NL63 spikes in a prefusion conformation in our

cryoET structures (Fig. 1), unlike SARS-CoV-2[18,19] and PEDV[31] virions, where both prefusion and postfusion spikes co-exist. The crown of the HCoV-NL63 spike displays a compact closed conformation with three RBDs down and RBMs hidden at the interface between domain A and domain B. Classification of cryoET data does not show any conformations other than the RBD-closed and domain 0 down conformation for HCoV-NL63 spike crown. This is different from the reported conformational transitions of prefusion spikes of other coronaviruses, like the transitions between standing and lying states of RBD in spikes of betacoronaviruses SARS-CoV[16,50], SARS-CoV-2[18–20] and MERS-CoV[16], between "swing out" and "proximal" states of domain 0 in the spikes of alphacoronaviruses PEDV[31] and CCoV-HuPn-2018[51], and expansion of S1 trimer in the spike of the other human common cold alphacoronavirus HCoV-229E[26].

This compact and stable structure of HCoV-NL63 spike may limit the access of glycan processing enzymes to the glycosylation sites, leading to the predominant high-mannose glycosylation of the HCoV-NL63 spike, in sharp contrast to the predominant complex glycan modifications of other human coronavirus spikes[33,34]. Our structure shows that high-mannose glycans form clusters on the surface of HCoV-NL63 spike (Fig. 3A, B) and protect epitopes critical for spike function. Our single particle cryoEM structure (PDB:7KIP) is resolved well enough to show not only the protein side chain densities but also the proximal portion of the glycan structure model (Fig. 2C, D). Epitope shielding by glycan caps is a well-known immune evasion strategy evolved in viruses. However, the glycan shield density depends on the choice of glycosylated protein model whether it is a static or a dynamic model with MD. In HCoV-NL63 spike, glycans at N844, N852, and N749 gate the S2' cleavage site on a neighboring protomer (Supplementary Fig. 7B), suggesting a tight regulation of protease cleavage at this site for fusion activation. No premature activation of the fusion peptide explains the lack of postfusion spikes on HCoV-NL63 virion in our subtomogram averages (Supplementary Fig. 2). HCoV-NL63 RBMs are not only hidden at the interface between domain A and domain B, but also protected by N358 and N506 linked glycans (Supplementary Fig. 7A). In order to interact with the receptor huACE2[52], HCoV-NL63 RBD needs to stand up to expose the RBM for receptor binding[53] although it is currently unknown how this conformational change is activated. Spike RBD is one of the most important neutralizing epitopes for coronaviruses and the hot spot of escape mutations[9,54]. While circulating seasonal HCoV-229E and OC43 undergo adaptive evolution in S1 at roughly half the rate of influenza A/H3N2 and a similar rate to influenza B viruses, HCoV-NL63 shows a lack of adaptive evolution[55] and the same genotype of HCoV-NL63 can re-infect people regularly[56]. This is at least partially attributed to the excellent epitope masking by the dense glycan shield (Fig. 3) on HCoV-NL63 spikes that prevents antibody binding and induction of strong host immune responses to place selection pressure on S gene.

Previous MD simulations have played a significant role in elucidating the mechanism of the spike crown and ACE2 interactions[30,57–63]. By integrating the in situ conformational landscape of prefusion spikes observed in cryoET with MD simulation of the entire spike tilting range here, we present a physiologically relevant analysis of dynamics of CoV spike ectodomain on the virus surface. There are a number of bending angles reported from prior simulations of the whole SARS-CoV2 spike, peaking between 10°–20° [24] and reaching 40° in a more recent study[64]. The discrepancy with our most probable observed bending angle (56°) can be attributed to the inadequacy of brute-force MD in capturing all the thermally accessible angles, which we overcome by introducing experimental information every 10° (Fig. 4A and Supplementary Fig. 10A). Highly bent spike conformations are also visualized in SARS-CoV2 virions, suggesting S trimers across multiple coronaviruses have a propensity to be stabilized in similar highly bent conformations under the observed packing densities[19,31].

The only conformational flexibility of native S ectodomain on HCoV-NL63 virions observed in our study resides in the long stalk, the striking feature of coronavirus viral glycoprotein. Tilting and twisting of spikes on the surface of virions were previously reported for coronaviruses SARS-CoV-2[18,19,24] and PEDV[31], as well as Env on HIV-1 virions and the purified ectodomain of Influenza hemagglutinin (HA)[18,65,66]. However, how the stalk dynamics could be regulated and how it correlates with viral infection are unknown. Our interaction energy computations of the clustered S stalk glycans revealed how global spike crown motions are affected by local contacts of N1242 and N1247 glycans precisely located at the hinge region. This adds mechanistic support to a previous structural observation of N-linked glycans at the junction between domains in a human glycoprotein mediating increased interdomain conformational flexibility[67]. The N-glycan-mediated optimal bending of the CoV spike further allows the most efficient shielding of the upper stalk region by the hinge glycans. This region is equivalent to the stem helix epitope targeted by cross-neutralizing antibodies against beta-coronaviruses. HCoV-NL63 N1242 sequon is highly conserved in both human and non-human CoVs across all four genera (Fig. 4B and Supplementary Fig. 10D) and the equivalent hinge glycans potentially modulate other CoV spike tilting in a similar way. This role of hinge glycans in regulating crown motions adds to the regulation of domain movement within the CoV crown by S1 glycans[29–31]. The link between HCoV-NL63 spike bending and epitope shielding, and virus infectivity warrants further investigation of how the S trimer motions affect CoVs infection and susceptibility to cross-neutralizing antibodies. Our discovery in the current study ultimately provides a rationale for developing therapeutics targeting stalk hinge glycosylation that can supplement existing anti-CoV therapeutics mainly targeting CoV protease and S RBD:receptor interactions.

The subtomogram average of the stalk is low-resolution, presumably caused by the conformational variability among the different stalks. Future analysis of larger numbers of tomograms and improved subtomogram averaging algorithm might yield higher resolution subtomogram average, which will assure the accuracy of the proposed model. There remain several limitations in the integrated molecular model that should also be noted. The transmembrane part of the spike is not considered experimentally or computationally. Hence, despite modeling the protein−glycan interactions with all-atom force fields suitable for both explicit and implicit simulations[68,69], residues of the S stalk nearest to the membrane had to be restrained during our simulations. The use of such additional restraints to avoid explicitly computing protein-membrane interactions can explain the absence of a transmembrane 'ankle' from our model that is seen in previous studies of SARS-CoV-2 spike[24]. This so-called ankle is also the least dynamic of the hinges and contributes minimally to the overall tilt of the crown as observed in the densities. Furthermore, as is often the case with MD, a longer simulation is expected to probe larger orientational deviations of the crown. Nonetheless, we have realized a key strength of data-guided MD here. Past integrative models e.g., of the ribosomes[70], would require non-equilibrium or steered MD to push structures between multiple known maps putting the biological relevance of the observed dynamics into question. Here, we have completely overcome the need to steer models between the cryoET maps, as equilibrium simulations are adequate for constructing a continuous distribution and capture functionally key interplay between protein-glycan and protein-protein interactions. Yet the space of azimuthal angles remains undersampled in MD relative to those seen in the cryo-EM images, given the very high diffusive barriers associated with these transformations. The azimuthal angle controls where in space the bending happens, while the tilt picks up the magnitude of this bending. Our cryo-EM data shows that the spikes are dispersed so that the inter-spike dynamics (and hence the azimuthal changes) does not affect spike bending. In an additional analysis (Supplementary Fig. 11), we

find that despite finite sampling there is little correlation between the azimuthal and tilt dynamics in the MD simulations. Due to the lower efficiency of HCoV-NL63 to grow in tissue culture than other human coronaviruses, HCoV-NL63 virus particles analyzed in this study were purified through density gradient ultracentrifugation that may have the potential to affect the number of spikes on the virus surface. Future study of HCoV-NL63 infected cells to investigate in situ assembled particles without virus isolation and purification is warranted. We demonstrated the involvement of hinge glycans in HCoV-NL63 spike-mediated virus entry using a pseudotype reporter virus that may present spikes differently from native HCoV-NL63. Future study of spike dynamics and virus infectivity of mutant HCoV-NL63 with hinge glycans modified is warranted.

## Methods

### Materials
4–15% Mini-PROTEAN® TGX™ Precast Protein Gels, 15-well, 15 µl (Bio-Rad); SimplyBlue™ SafeStain (Invitrogen); dithiothreitol (Sigma); iodoacetamide (Sigma); alpha lytic protease (New England BioLabs); chymotrypsin (Athens Research and Technology); AspN (Promega); Glu-C (Promega); trypsin (Promega); endoglycosidase H (Promega); PNGaseF (Promega); 18O water (Cambridge Isotope Laboratories).

### Cell culture and virus
MA104 (ATCC CRL-2378.1) cells were maintained at 37 °C in a fully humidified atmosphere with 5% $CO_2$ in M199 (Gibco) medium, hamster fibroblast cell line BHK21 (ATCC CCL-10) and adenovirus 5 DNA transformed human kidney epithelial cell line 293 c18 (ATCC CRL-10852) were maintained at 37 °C in a fully humidified atmosphere with 5% $CO_2$ in DMEM (Gibco). All culture media were supplemented with penicillin and streptomycin and 10% FBS (Hyclone). HCoV-NL63 was obtained from BEI Resources, NIAID, NIH: Human Coronavirus, NL63, NR-470.

### Virus production and purification
HCoV-NL63 was produced and purified as described previously[10]. Briefly, culture supernatants of HCoV-NL63 infected cells were harvested when clear cytopathic effect (CPE) developed. The virus was pelleted from clarified culture supernatants clarified through 0.2 µm filter by centrifugation through a 20% sucrose cushion at 72,000 × g for 2 h at 4 °C with a SW28 rotor. Virus was resuspended with NTE buffer (20 mM Tris, pH 8.0, 120 mM NaCl, 1 mM EDTA) overnight at 4 °C before ultracentrifugation in 20% OptiPrep (Sigma-Aldrich) at 360,000 × g for 3.5 h 4 °C with a NVT65.2 rotor. The virus band was extracted, and buffer exchanged to the NTE buffer using an Amicon Ultra-2 Centrifugal Filter Unit with Ultracel-100 membrane (Millipore) at 4 °C and processed immediately for cryoET. Purified virus was subjected to electrophoresis with 4–12% sodium dodecyl sulfate-polyacrylamide gel (Invitrogen) followed by Coomassie-staining with SimpleBlue SafeStain (Invitrogen). To demonstrate glycosylation of spikes, purified virus was digested with PNGaseF (New England Biolabs) followed by gel analysis.

### HCoV-NL63 spike pseudotyped VSV-Luc reporter virus production
BHK21 cells were transfected with HCoV-NL63 S expression plasmid prior to infection with VSV-G pseudotyped ΔG-luciferase VSV at MOI 3.0. Six hours post infection, cells were washed with PBS and fresh medium containing anti-VSV-G hybridoma was added to neutralize the residual VSV-G pseudovirus. Culture supernatant was harvested after 24 h of incubation, clarified by centrifugation and stored at −80 °C in aliquots. To make virus pseudotyped with HCoV-NL63 spike with specific N-linked glycan removed, a similar protocol was used except cells were transfected with plasmids expressing mutant HCoV-NL63 S. Site-directed mutagenesis was used to mutate specific Asn to Asp in the HCoV-NL63 S expression plasmid. The primer pairs to make ΔglycN1242, ΔglycN1247 and ΔglycN1242;ΔglycN1247 HCoV-NL63

spike expression plasmids are 5′-CTCCTTTTGATTTAACATATCTTAA TTTGAG-3′/5′-TATGTTAAATCAAAAGGAGTCAAGTCAAAATT-3′, 5′-CA TATCTTGATTTGAGTTCTGAGTTGAAGC-3′/5′-GAACTCAAATCAAGAT ATGTTAAATTAAAAGG-3′, and 5′-TTGATTTAACATATCTTGATTTGAG TTCTGAGTTGAAGC-3′/5′-AAATCAAGATATGTTAAATCAAAAGGAGTC AAGTCAAAATT-3′, respectively.

### CryoET sample preparation, data acquisition, image processing and subtomogram averaging
Three microliters of purified HCoV-NL63 virus particles (4.5 mg/mL) with 5 nm gold fiducials were applied to 300-mesh R2/1 + 2 nm C-film grids (Quantifoil) to prepare grids for Cryo-EM analysis as we previously described[10]. Cryo-ET data were collected by loading frozen grids into a Thermo Fisher Talos Arctica transmission electron microscope operated at 200 kV and recording images on a Gatan Summit K2 detector in counting mode with a 20 eV energy filter in zero-loss mode. Tomographic tilt series were collected between +50° and −50° using a bi-directional scheme starting at +21° with a 2° angular increment using SerialEM software[71]. The total dose applied to each tilt series was 120 e⁻/Å² with equal distribution among the 51 tilt images. The nominal magnification was 53,000×, with corresponding pixel size of 2.20 Å. Defocus for each tilt series varied between −2 µm and −5.5 µm. Frames of each tilt angle movie stack were motion-corrected using MotionCor2 software and tilt series compiled using EMAN2 *e2import.py* build virtual tilt stacks command. Tilt images were automatically aligned and reconstructed into 3D volumes using EMAN2 software[72]. More than 160 tomograms were judged as sufficient for further analysis based on average alignment loss of reconstructed tomograms (see Supplementary Table 1).

All processing for subtomogram averaging was performed in EMAN2. 18,356 individual spike particles were picked automatically using the neural network boxer, CTF-corrected and then extracted into 3D subvolumes. An initial model was generated directly from 50 high SNR subvolumes. This model roughly matched our previous single particle cryoEM structure of the S crown (EMD:22889) at low resolution. Using all particles and applying C3 symmetry, a refined map of the S crown and upper stalk was achieved at 6.9 Å resolution after iterative subtomogram and subtilt alignment (see Supplementary Table 1). As described in the main text, the crown region was well-resolved in the refined map with poorly resolved upper stalk density and without any resolved density corresponding to the viral membrane. A spherical 3D mask was generated in the expected region of the stalk and membrane (Supplementary Fig. 1), and iterative focused classification was performed while releasing the C3 symmetry. The focused classification yields spikes with membranes at different tilting angles. Then, particles from different classes are re-aligned so that the membrane and stalk are at roughly the same pose, and the spikes are tilted in different angles. Averaging all particles together at this orientation yields the initial structure of the stalk. Subsequent orientation refinement is then performed on all subvolumes starting from this pose and initial model, focusing on the stalk region and imposing C3 symmetry. This resulted in a moderately resolved structure of the viral membrane with stalk region at 11.5 Å resolution. Finally, we compare the difference of orientation assignment of the spike and stalk for each individual particle, and decompose the relative angle to two axes, i.e., bending and in-plane rotation. Particles were then manually clustered into bins based on either bending or in-plane rotation angle and averaged to generate 3D class averages.

Visualization and model docking into the density maps was performed using UCSF Chimera and ChimeraX softwares using the fit-in-map tool[73,74].

### Isolation of HCoV-NL63 S protein
Approximately 2440 µg of total viral protein from HCoV-NL63 particles was resolved by SDS-PAGE gel (Bio-Rad) and stained with

SimplyBlue™ (Invitrogen). The amount of S protein was estimated to be 10% of the total protein based on the intensity of the stained bands. Bands at 250 kDa (S protein) were cut and then combined to make 10-µg aliquots of S protein.

## Analysis of site-specific N-linked glycopeptides for HCoV-NL63 S protein by LC−MS

Eight 10-µg aliquots of HCoV-NL63 S protein were reduced by incubating the gel bands with 10 mM of dithiothreitol (Sigma) at 56 °C and alkylated by 27.5 mM of iodoacetamide (Sigma) at room temperature in dark. The eight aliquots were then digested respectively using alpha lytic protease (New England BioLabs), AspN (Promega), chymotrypsin (Athens Research and Technology), Glu-C (Promega), trypsin (Promega), a combination of trypsin and chymotrypsin, a combination of chymotrypsin and AspN, or a combination of trypsin and AspN. The resulting peptides were separated on an Acclaim™ PepMap™ 100 C18 column (75 µm × 15 cm) and eluted into the nano-electrospray ion source of an Orbitrap Eclipse™ Tribrid™ mass spectrometer (Thermo Scientific) at a flow rate of 200 nL/min. The elution gradient consists of 1−40% acetonitrile in 0.1% formic acid over 370 min followed by 10 min of 80% acetonitrile in 0.1% formic acid. The spray voltage was set to 2.2 kV and the temperature of the heated capillary was set to 275 °C. Full MS scans were acquired from m/z 200 to 2000 at 60k resolution, and MS/MS scans following higher-energy collisional dissociation (HCD) with stepped collision energy (15%, 25%, 35%) were collected in the orbitrap at 15k resolution. pGlyco v3.0[75] was used for database searches with mass tolerance set as 20 ppm for both precursors and fragments. The database search output was filtered to reach a 1% false discovery rate for glycans and 10% for peptides. The glycan and peptide assignment for each spectra was then manually validated after filtering. Quantitation was performed by calculating spectral counts for each glycan composition at each site. Any N-linked glycan compositions identified by only one spectra were removed from quantification. N-linked glycan compositions were categorized into 19 classes (including Unoccupied): HexNAc(2)Hex(9-5)Fuc(0-1) was classified as M9 to M5 respectively; HexNAc(2)Hex(4-1)Fuc(0-1) was classified as M1-M4; HexNAc(3-6)Hex(5-9)NeuAc(0-1) was classified as Hybrid with HexNAc(3-6)Hex(5-9)Fuc(1-2)NeuAc(0-1) classified as F-Hybrid; Complex-type glycans are classified based on the number of antenna and fucosylation: HexNAc(3)Hex(3-4)Fuc(0)NeuAc(0-1) is assigned as A1 with HexNAc(3)Hex(3-4)Fuc(1-2)NeuAc(0-1) assigned as F-A1; HexNAc(4)Hex(3-5)Fuc(0)NeuAc(0-2) is assigned as A2/A1B with HexNAc(4)Hex(3-5)Fuc(1-5)NeuAc(0-2) assigned as F-A2/A1B; HexNAc(5)Hex(3-6)Fuc(0)NeuAc(0-3) is assigned as A3/A2B with HexNAc(5)Hex(3-6)Fuc(1-3)NeuAc(0-3) assigned as F-A3/A2B; HexNAc(6)Hex(3-7)Fuc(0)NeuAc(0-4) is assigned as A4/A3B with HexNAc(6)Hex(3-7)Fuc(1-3)NeuAc(0-4) assigned as F-A4/A3B; HexNAc(7)Hex(3-8)Fuc(0)NeuAc(0-1) is assigned as A5/A4B with HexNAc(7)Hex(3-8)Fuc(1-3)NeuAc(0-1) assigned as F-A5/A4B.

## Analysis of deglycosylated HCoV-NL63 S protein by LC−MS

Six 10-µg aliquots of SARS-CoV-2 S protein were reduced by incubating the gel bands with 10 mM of dithiothreitol (Sigma) at 56 °C and alkylated by 27.5 mM of iodoacetamide (Sigma) at room temperature in dark. The six aliquots were then digested respectively using AspN (Promega), chymotrypsin (Athens Research and Technology), Glu-C (Promega), trypsin (Promega), a combination of chymotrypsin and AspN, or a combination of trypsin and AspN. Following digestion, the extracted peptides were deglycosylated by Endoglycosidase H (Promega) followed by PNGaseF (Promega) treatment in the presence of 18O water (Cambridge Isotope Laboratories). The resulting peptides were separated on an Acclaim™ PepMap™ 100 C18 column (75 µm × 15 cm) and eluted into the nano-electrospray ion source of an Orbitrap Eclipse™ Tribrid™ mass spectrometer (Thermo Scientific) at a flow rate of 200 nL/min. The elution gradient consists of 1−40% acetonitrile in 0.1% formic acid over 370 min followed by 10 min of 80% acetonitrile in 0.1% formic acid. The spray voltage was set to 2.2 kV and the temperature of the heated capillary was set to 275 °C. Full MS scans were acquired from m/z 200 to 2000 at 60k resolution, and MS/MS scans following collision-induced dissociation (CID) at 38% collision energy were collected in the ion trap. The spectra were analyzed using SEQUEST (Proteome Discoverer 2.5, Thermo Fisher Scientific) with mass tolerance set as 20 ppm for precursors and 0.5 Da for fragments. The search output was filtered to reach a 1% false discovery rate at the protein level and 10% at the peptide level. The site assignment for each spectrum was then manually validated after filtering. Occupancy of each N-linked glycosylation site was calculated using spectral counts assigned to the 18O-Asp-containing (PNGaseF-cleaved) and/or HexNAc-modified (EndoH-cleaved) peptides and their unmodified counterparts.

## Quantification and statistical analysis

Raw glycoproteomic data from the mass spectrometers was searched using SEQUEST (Proteome Discoverer 2.5) and pGlyco3. Search results from SEQUEST were filtered to reach a 1% false discovery rate at the protein level and 10% at the peptide level; search results from pGlyco3 were filtered to reach a 1% false discovery rate at the glycan level and 10% at the peptide level. All spectral assignments were manually validated after applying false discovery rate filtering.

## Fully glycosylated HCoV-NL63 spike modeling

The NL63 spike models previously solved with cryoEM (PDB:7kip, PDB:5szs) already contain glycan molecules attached to ASN residues on the surface of the spike; these were previously modeled to fit the resolved cryoEM density. We further added glycan molecules at each site to create full glycan models, even though these may not have been resolved in the density due to their flexibility, for the purpose of MD simulation, glycan shield analysis, and visualization. These glycans were added using the SegMod tool which is part of the Chimera plugin Segger v2.9.2 as follows: (1) the model of each saccharide was generated using the Elbow tool in Phenix[76], and positioned into the model with proper connecting bond lengths and angles, (2) the dihedrals of the new bonds were adjusted automatically to minimize clashes between the added glycans and existing atoms in the entire spike model. Three different glycan types were added at each site, depending on the highest population at that site: high-mannose, hybrid, or complex. After adding all glycans, the entire model was refined with phenix.real_space_refinement. In the latter, each glycan moved minimally (less than 1 Å RMSD), indicating the initial placements created reasonable models. This process was applied to the HCoV-NL63 spike model (PDB: 7kip), SARS-CoV2 spike models with stalks[30] (based on PDB:6vyb), and other spike models: HKU1 (PDB:5i08), SARS (PDB:5 × 58), and MERS (PDB:5 × 59).

## Number of surface residues accessible by fab

This number reflects the number of amino acid residues accessible by a probe roughly the size of the complementarity-determining regions (CDRs) of a Fab fragment of an antibody. It is calculated by testing whether a sphere with the radius of 20 Å can be placed such that it contacts a given residue but no other protein residue or saccharide in a glycan. This is calculated for each residue in a virus spike and illustrated in Supplementary Fig. 8. This method is also automated and available in the SegMod Chimera plugin (v2.9.2).

## HCoV-NL63 stalk modeling

We used I-TASSER protein folding software[35] to fold a monomer of the extracellular region of HCoV-NL63 S stalk (residues 1216–1297). The highest confidence model indicated that a monomeric HCoV-NL63 S stalk likely consisted of a short alpha helix (residues 1216–1228) followed by a disordered region (residues 1229–1245) that then transitioned into a longer alpha helix (residues 1246–1297), similar to other

coronavirus spike protein models[25,77] (Supplementary Fig. 7). Three monomeric S stalk models were then manually grafted onto the full atomic crown model via the overlap region of the short alpha helix (residues 1216–1224) from the previous single particle model (PDB: 7KIP) using Visual Molecular Dynamics (VMD) version 1.9.3[78] to construct the entire trimeric spike assembly. Flexible fitting of the trimer model into the cryoET density and subsequent energy minimization using Molecular Dynamics Flexible Fitting (MDFF) reoriented the stalk monomers, improving their inter-helix contacts. Details of the MDFF parameters are provided below. This reduced the solvent-exposed hydrophobic surface area by 36% (47 Å² → 30 Å²) in the region encompassing residues 1224–1297. However, this model still possessed some contentious residue orientations in the lower coiled-coil region (residues 1245–1297) with buried glycosylation sites.

To improve the above stalk model, we then generated an Alpha-Fold prediction of the full trimeric spike ectodomain[79]. Both MDFF-optimized I-TASSER and AlphaFold models showed high agreement in the structure of the upper coiled-coil region (residues 1216–1245) (Supplementary Fig. 6A). However, the lower coiled-coil region (residues 1280–1297) had a low AlphaFold confidence score (Supplementary Fig. 6B), matching steric issues with the ITASSER-MDFF model. We then remodeled the lower coiled-coil region of the spike trimer using the CCbuilder 2.0 program[80]. This tool is parameterized against the well-established configuration of coiled-coil to build an idealized multimer, and concomitantly uses a classical MD force field to optimize helix register, inter-helix orientation and promote simulation stability. The CCbuilder parameterized coiled-coil was attached back to the complete model starting at position 1245, replacing the region modeled by the initial flexible fitting (residues 1242–1297). The N1242 and N1247 glycans on the stalk were introduced into the model by using the CHARMM-GUI interface[37], and chosen based on agreement between our mass spectrometry assignments and a previous one[11].

The different spike protein bending angles observed in the CryoET experimental data were recreated by fitting the complete model to seven different density maps, each capturing an observed bending angle. To this end, first, the ChimeraX visualization tool[81] was used to align the lower stalk region of HCoV-NL63 protein model so that the base of the lower coiled-coil region was adjacent to the viral membrane. The complete model was then fitted to the experimental density maps using the Molecular Dynamics Flexible Fitting (MDFF) method[82]. All MDFF simulations were performed using the molecular dynamics simulation (MD) software, NAMD 2.13[49], and the CHARMM36 force field[83]. During the MDFF simulations, a potential energy function ($U_{EM}$), obtained from the cryoET density map, was applied to all Cα outside of the unstructured region (residues 1224–1241) to bias the protein into adopting the experimentally resolved bending angle using a g-scale (i.e., map-model coupling constant) of 1. To avoid unwanted perturbation of the model, restraints were added to maintain cis-peptide, secondary structure, and chirality of the protein. Additionally, the overall structure of the crown and the coiled-coil regions were preserved using a domain restraint in the form of a RMSD bias being applied to the Cα of all residues not in the unstructured region using NAMD Targeted Molecular Dynamics module[49]. Finally, excessive flexibility of the lower stalk region due to the lack of transmembrane region was avoided by placing a positional restraint on the C-terminal of the protein. Following the initial MDFF simulations, the unstructured region was refined by performing four sequential MDFF simulations. In each simulation, the g-scale of the MDFF potential applied to all protein Cα atoms was reduced from 0.3 to 0 by increments of 0.1. This procedure resulted in seven fitted models with each representing a different experimentally resolved bending angle.

## Molecular dynamics simulation

All atom explicit solvent simulations of the seven different spike protein models were performed using NAMD and CHARMM36 force fields

(see Supplementary Table 2). Each model was solvated in a large water box, permitting large bending angles, with the dimensions of 429 Å × 319 Å × 429 Å and sodium and chloride ions were added to ensure a neutral system charge. The final solvated system was approximately 5.8 million atoms. A constant temperature of 310 K was maintained using Langevin dynamics with a damping coefficient of 1 ps⁻¹, and constant pressure was maintained at 1 atm using the Langevin piston method. Long range electrostatic forces were computed using the particle mesh Ewald method, while Van der Waals and short-range electrostatics were smoothly truncated at 12 Å with a switch function implemented at 10 Å. To allow larger timesteps before force integration, hydrogen mass repartitioning was used. The initial structures were minimized using the default conjugate gradient energy minimization algorithm implanted in NAMD for 11,000 steps and the system was equilibrated for 10 ns. Following equilibration, the structures were simulated in triplicates for ~50 ns with forces being integrated every 4 fs with a positional restraint on the C-terminal. To capture large movements that would be otherwise inaccessible in explicit simulations[69], ~100 ns of implicit solvent simulations were performed on the final frames of each explicit simulation. Implicit solvent simulations were performed using the Generalized Born Implicit solvent method implemented in NAMD. Simulations were performed using a solvent dielectric of 80 and an ion concentration of 0.1 M. The Born radius cutoff parameter was set to 14 Å with the switch distance and cutoff set to 15 Å and 16 Å respectively. This resulted in a total sampling time of 350 ns per map with a total of 2.45 μs. For simulations investigating the effects of glycan modulation, glycans were removed from the final frames of the explicit simulation and simulated for 100 ns in an implicit solvent.

## MD trajectory analysis

The bending of the crown relative to the stalk was determined by finding a vector that passed through the center of the lower stalk region while remaining normal to the virion surface and a second vector that was defined as running through the center of the HCoV-NL63 crown. Shown in Supplementary Fig. 9, these vectors also represent the dominant principal axis of the crown and the stalk respectively. The bending angle was then defined as the arccosine of the cross product between these two vectors. The ASA measurements were performed using VMD. Each ASA measurement was calculated with respect to the specified spike epitopes using a probe of radius 20 Å, which mimes a CDR region of an antibody. Note, this maximum ASA represents the largest area on the probe that is accessible to the protein, and not the vice versa i.e., area of the protein that is accessible to the probe - a measure traditionally used to monitor the solvent accessible surface. Consequently, huge cavities left in the structures e.g., due to the deletion of multiple glycans will be dramatically more accessible to the probe and shielded spaces. Hinge glycan contacts with HCoV-NL63 stalk protein residues were defined as the minimum pairwise distance between hinge glycan atoms and the specified HCoV-NL63 stalk protein residue atoms. In the case of the wild-type HCoV-NL63 spike simulations, minimum hinge contact distance was defined as the minimum pairwise distance between atoms from either one of the hinge glycans (on N1242 or N1257) and the upper stalk protein residues 1216–1230. The interaction energy of the hinge glycans (N1242,1247) were measured using the NAMD energy plugin[49]. Energy was either measured from both hinge glycans to the rest of the system or with respect to individual hinge glycans.

The simulations starting from the 70° bend had 73.5% of the trajectory going towards lower angles, and only 26.5% going to higher bent conformations. Furthermore, Of the trajectories with bending angles transitioning above 70°, only 7% (out of the aforementioned 26.5%) reached all the way to 80° (1.8% of all simulations starting at 70° or 0.2% of all measured frames). None of the trajectories starting from other angles (i.e. <70°) make it to 80°. On the other end of the bending

range, 5.8% of all trajectories reached less than 10° (i.e., 10°–0°) which is still an order of magnitude more than that observed over the range from 70° to 80°. So, our simulations encompassed extremities of bent conformations, even without starting from a low-probability model (i.e., 80°).

## Reporting summary

Further information on research design is available in the Nature Portfolio Reporting Summary linked to this article.

## Data availability

The cryoEM subtomogram average map of the crown reported in this study is deposited in the Electron Microscopy Data Bank (EMDB) under accession code EMDB-29395; Atomic model of spike protein and glycans is deposited to wwProtein Data Bank (PDB) under accession code PDB:8FR7. Models previously reported and referenced in this work are publicly available with the PDB accession codes PDB:7KIP, PDB:5SZS, 6VYB, PDB:5I08, PDB:5X58 and PDB:5Z59 [https://doi.org/10.2210/pdb5X59/pdb].The mass spectrometry data have been deposited to the ProteomeXchange Consortium via the PRIDE partner repository with the dataset identifier PXD039247. Source data are provided with this paper.

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

## Acknowledgements

We thank Drs. Corey Hecksel and Patrick Mitchell for expert support of Stanford-SLAC Cryo-EM Center and the SLAC National Accelerator Laboratory to conduct these studies during a university-wide pandemic shutdown. We thank support from DOE Office of Science through the National Virtual Biotechnology Laboratory, a consortium of DOE national laboratories focused on response to COVID-19, with funding provided by the Coronavirus CARES Act and National Institutes of Health grants (R01AI148382 to W.C.; R21MH125285 to M.C.; R01GM080139 to M.C. via Steven Ludtke at Baylor College of Medicine). The work that used mass spectrometry was supported by the National Institutes of Health grant (R01GM130915 to L.W.). A.S. acknowledges the CAREER award from NSF

(MCB-1942763), National Institute of Neurological Disorders and Stroke (RO1NS119505) and AstraZeneca. The simulation work used the Extreme Science and Engineering Discovery Environment (XSEDE), which is supported by National Science Foundation grant number ACI-1548562, and Oak Ridge Leadership Computing Facility, supported by the Office of Science, Department of Energy (DE-AC05-00OR22725).

## Author contributions

W.C., A.S. and J.J. designed and supervised the study. J.J. provided purified HCoV-NL63 samples and performed all the virology experiments. P.Z. and L.W. performed glycosylation analysis by mass spec. D.C. collected cryoET data and analyzed it with M.C. E.W. and A.S. performed molecular dynamic simulations. G.P. generated molecular models and performed Q-score analysis. D.C., E.W., M.F.S., G.S. J.J., A.S. and W.C. wrote the manuscript together.

## Competing interests

The authors declare no competing interests.
