## [Peer Review File · Nature Communications]

REVIEWER COMMENTS

Reviewer #1 (Remarks to the Author):

HCoV-NL63 is an endemic seasonal human coronavirus which shares the same huACE2 receptor with SARS-CoV-2. Compared to SARS-CoV-2, HCoV-NL63 shows a lack of adaptive evolution, and is more glycosylated on its spikes. In this manuscript, Chmielewski et al. have integrated multiple experimental and computational approaches to discover and demonstrate the impact of a single glycan on the dynamics and infectivity of HCoV-NL63. The authors have imaged unfixed virions by cryo-ET and analyzed its spike protein structures and statistics. Combined with MS, the glycan compositions were analyzed and an atomic model of the most abundant glycan topology was built into the density of their previous single particle cryoEM map, giving them the opportunity in analyzing the glycan shielding. Atomic models of the ectodomain were built for glycosylated spike resolved at tilt angles of 10o, 20o...70o, which were subjected to MD analysis. The simulation has revealed the bending profile of the spike, revealing that two N-glycans (N1242 & 1247) near this disordered hinge region may have important roles in spike tilting. Further MD and infectivity assay using a VSV pseudovirus have revealed a role of the conserved N-linked glycan N1242 in both spike bending and infectivity.

Overall, the discoveries presented in this work is novel, and the logic among various experimental approaches is fluent. The authors have also discussed the limitations in this work, such as uncertainties in building the atomic model for MD and short simulation duration, which is helpful for readers. My main concerns locate at the sample preparation procedure and infectivity assay using the VSV pseudovirus. The manuscript needs major revision before being recommend for publication in Nature Communications.

Major points:

1. The authors pelleted the unfixed virions through 20% sucrose cushion followed by OptiPrep gradient, then filtered the virions through a membrane filter to exchange the buffer. The pelleting and filtering raise concerns on the intactness of the virions, as the thermal stability of coronaviruses and their spike proteins are known to be weak (especially when they are not chemically fixed). Also, for how long and at what temperature was the virion sample stored during each step before plunge-freezing? This is not clearly written in the method part on virus preparation. These possibly result in only ~9 spikes (according to line 130) found per virion and some virions look bald (fig.1A), this number is significantly less than that of SARS-CoV-2 and PEDV (~30 spikes/virion). This suggests that the virions are possibly damaged during sample preparation. If so, the average distance between neighboring spikes is over-estimated, and this will impact the spike dynamics analysis, since the spikes can contact each other on an intact virion. The authors shall analyze this possibility and provide a control for the virion intactness by cryo-ET of freshly prepared & unconcentrated HCoV-NL63 in the supernatant.

2. The cryo-ET data processing part is not clearly stated. Detailing this part in the M&M session will not only justify the robustness of their structures, but also help the readers in learning cryo-ET:

(1) In the M&M part, the authors wrote: “approximately 15,000 individual spike particles were manually picked... Subtomogram averaging was then performed using ~10,000 particles”. However, in line 133, they wrote “Subtomogram averaging analysis of 18,356 spikes”. The numbers are inconsistent.

(2) What software did they use for subtomogram averaging? How was the “focused classification” (as stated in line 153) done?

(3) A supplementary table containing detailed information on cryo-ET data acquisition and processing must be provided.

3. Why is the cryo-ET determined spike with a bending angle of 80° not modeled for MD?

4. The influence of the hinge glycan N1242 on infectivity was analyzed using VSV-Luc reporter pseudovirus. While coronaviruses bud in ERGIC, where N-glycosylation gets further modified, VSV buds at the plasma membrane. In addition, coronaviral E protein has a function of intracellular retention, which helps concentrating viral structural proteins at the ERGIC to favor viral particle assembly. Therefore, the difference in budding and the missing E from the VSV pseudovirus can result in differences in glycosylation. The authors shall analyze the glycan compositions of the VSV pseudovirus, especially for the N1242 and N1247 sites, to control for the possible differences in glycosylation.

Minor points:

1. Figure 4D is never mentioned in the manuscript.

2. Fig. S2B is not mentioned in the main text. There is no discussion on why/how two spikes are connected in the stalk.

3. Fig. 1G: the angles are boxed in colors, however, the colors are not explained.

4. Adding a comparison animation of the MD simulation of S-trimers with and without the N1242 glycan to the manuscript will be highly appreciated.

5. I’m wondering if the most favorable spike tilt 56° is the optimal tilt conformation, what would be the virological implications of this angle? Would the spike mediate receptor binding better so as to influence viral infectivity? A little discussion expansion on correlation between the spike tilting and viral infectivity will be appreciated.

Reviewer #2 (Remarks to the Author):

The manuscript by Chmielewski et al. provides important knowledge on the molecular dynamics of virion-bound HCoV-NL63 spike glycoprotein (S) and proposes that a specific N-glycan within a flexible hinge region at the stalk-crown interface confers some of the conformational flexibility. By applying

focused classification, the authors were able to resolve the otherwise orientationally flexible stalk domain, indiscernible at the complete cryo-EM map. By further isolating image classes with various combinations of crown and stalk domain orientations, the authors also elegantly demonstrate highly variable tilting angles of virion-bound spike, with the crown domain not only tilting, but also rotating relative to the stalk domain, with this flexibility imposed by the hinge region. Intriguingly, in silico removal of the hinge-resident N-glycan at N1242, or all the S glycans, results in a shift of most energetically favorable tilting angles from 56° to around 25°, decreasing S flexibility, and possibly increasing exposure to antibodies. Moreover, HCoV-NL63 S-pseudotyped VSVdeltaG with said N-glycan deletion exhibits substantially decreased infectivity. The work provides extremely valuable insight into how glycans affect the conformational stability of proteins – a subject that is still underexplored. The authors provide compelling evidence for plausible roles of glycans in HCoV-NL63 biology, which could be further strengthened by in vitro experiments. The glycoproteomic analysis of virion-derived S is a welcome addition for determining predominant site-specific glycan compositions, however, a few details should be clarified.

1. The mass spectrometry sample and data analysis approach should be clarified. The multiple digests are typically employed to enhance coverage, and it is not unusual to pool them prior to analysis. Is that what was performed here, or were they analyzed separately? Given the semi-quantitative nature of spectral counting, were any measures taken to estimate variation (biological or technical replicates)?

2. The increased access to antibody molecules upon removal of N-glycans in the flexible hinge region makes sense, as suggested by the molecular modelling experiments (Figure 4C, 4D). However, the software-determined approach measuring the accessible area of the probe, and not of the exposed spike area is somewhat counterintuitive. It would strengthen the manuscript, if an in vitro approach was employed to further test the validity of the claims. For example, suitable neutralizing antibodies, or sera from seropositive individuals could be used to measure antibody binding capacity, or neutralization of pseudotyped virions.

3. Figure 5C. The plot elements should be described in the legend (measure of center, error bars, and statistical test). If the data is normalized to the infectivity of wild type, why are there error bars on the wild type data? What is the reproducibility of the data? Pseudovirus infectivity data can be rather variable, and the information on numbers of independent experiments is missing.

Minor:

Lines 254-255. The term “monosaccharide” should be used instead of “glycan”.

Line 319. It would be useful to mention exact numbers for glycan shield density for the other coronaviruses.

Lines 484-485. How was the “conservation of glycosylation” determined as opposed to amino acid conservation (previous sentence)? Is it based on experimental data from the literature? If so, how many and which viruses? Or is it algorithm-based prediction?

Figure S7D. Please add amino acid residue numbers for the start/end of the shown ranges.

Responses to Reviewers Comments (blue font)

REVIEWER COMMENTS

Reviewer #1 (Remarks to the Author):

HCoV-NL63 is an endemic seasonal human coronavirus which shares the same huACE2 receptor with SARS-CoV-2. Compared to SARS-CoV-2, HCoV-NL63 shows a lack of adaptive evolution, and is more glycosylated on its spikes. In this manuscript, Chmielewski et al. have integrated multiple experimental and computational approaches to discover and demonstrate the impact of a single glycan on the dynamics and infectivity of HCoV-NL63. The authors have imaged unfixed virions by cryo-ET and analyzed its spike protein structures and statistics. Combined with MS, the glycan compositions were analyzed and an atomic model of the most abundant glycan topology was built into the density of their previous single particle cryoEM map, giving them the opportunity in analyzing the glycan shielding. Atomic models of the ectodomain were built for glycosylated spike resolved at tilt angles of 10o, 20o...70o, which were subjected to MD analysis. The simulation has revealed the bending profile of the spike, revealing that two N-glycans (N1242 & 1247) near this disordered hinge region may have important roles in spike tilting. Further MD and infectivity assay using a VSV pseudovirus have revealed a role of the conserved N-linked glycan N1242 in both spike bending and infectivity. Overall, the discoveries presented in this work is novel, and the logic among various experimental approaches is fluent. The authors have also discussed the limitations in this work, such as uncertainties in building the atomic model for MD and short simulation duration, which is helpful for readers. My main concerns locate at the sample preparation procedure and infectivity assay using the VSV pseudovirus. The manuscript needs major revision before being recommend for publication in Nature Communications.

Major points:

1. The authors pelleted the unfixed virions through 20% sucrose cushion followed by OptiPrep gradient, then filtered the virions through a membrane filter to exchange the buffer. The pelleting and filtering raise concerns on the intactness of the virions, as the thermal stability of coronaviruses and their spike proteins are known to be weak (especially when they are not chemically fixed). Also, for how long and at what temperature was the virion sample stored during each step before plunge-freezing? This is not clearly written in the method part on virus preparation. These possibly result in only ~9 spikes (according to line 130) found per virion and some virions look bald (fig.1A), this number is significantly less than that of SARS-CoV-2 and PEDV (~30 spikes/virion). This suggests that the virions are possibly damaged during sample preparation. If so, the average distance between neighboring spikes is over-estimated, and this will impact the spike dynamics analysis, since the spikes can contact each other on an intact

virion. The authors shall analyze this possibility and provide a control for the virion intactness by cryo-ET of freshly prepared & unconcentrated HCoV-NL63 in the supernatant.

We thank the reviewer for pointing out the missing details of virus preparation. We have added a detailed description about temperature/time during virus purification in the Materials and Methods. Our previous draft may have led to a possible misinterpretation of the number of virions in our tomograms from which spikes were actually analyzed by subtomogram averaging. We returned to the data and determined the number of spikes per particle by manual identification of spikes within tomograms. In our purified HCoV-NL63 virion, there are 20 ± 13 spikes/HCoV-NL63 virions in a selected pool of 154 intact virus particles. This number is similar to what is reported for spikes in prefusion conformation on SARS-CoV-2 virion: 23 ± 9 in (Ke et al. 2020) and 26 ± 15 in (Yao et al. 2020), but lower than ~ 40 spikes per virion reported in (Turoňová et al. 2020). Unlike betacoronaviruses, HCoV-NL63 spike is not cleaved between S1 and S2, and the spike displays only one closed prefusion conformation observed in both purified soluble spike proteins (Walls et al. 2016) and in situ on purified virions (Zhang et al. 2020). On SARS-CoV-2 virus particles, not only prefusion but also postfusion spikes were observed, and the prefusion spikes display both open and closed conformations. The high flexibility of the SARS-CoV-2 spikes may explain the loss of prefusion spikes in open conformation after centrifugation through sucrose cushion as reported in (Ke et al. 2020). The purity of our HCoV-NL63 virus prep and stability of HCoV-NL63 spike are shown in the coomassie gel in new Figure S3B, appearing to be more stable than betacoronavirus spikes.

2. The cryo-ET data processing part is not clearly stated. Detailing this part in the M&M session will not only justify the robustness of their structures, but also help the readers in learning cryo-ET:

(1) In the M&M part, the authors wrote: “approximately 15,000 individual spike particles were manually picked... Subtomogram averaging was then performed using $\sim 10,000$ particles”. However, in line 133, they wrote “Subtomogram averaging analysis of 18,356 spikes”. The numbers are inconsistent.

We have revised our manuscript to clarify the data processing method. The same dataset was in fact processed twice using different versions of the EMAN2 software package resulting in different numbers of spikes. The results shown in the paper are from the final version, but some of the numbers in the original paper were not updated and thus incorrect. We have now updated the numbers in the main text and the method section. Our final analysis for subtomogram averaging included 18,356 spikes in both intact and apparently broken virions.

(2) What software did they use for subtomogram averaging? How was the “focused classification” (as stated in line 153) done?

We revised the method of subtomogram averaging and provided a supplemental figure (new Figure S1) showing the processing details of focused classification.

(3) A supplementary table containing detailed information on cryo-ET data acquisition and processing must be provided.

As suggested, we provided a supplemental table showing detailed information on cryoET data acquisition and processing.

3. Why is the cryo-ET determined spike with a bending angle of 80° not modeled for MD?

The simulations starting from the 70° bend had 73.5 % of the trajectory going towards lower angles, and only 26.5 % going to higher bent conformations. Furthermore, Of the trajectories with bending angles transitioning above 70° , only 7% (out of the aforementioned 26.5%) reached all the way to 80° (1.8% of all simulations starting at 70° or 0.2% of all measured frames). None of the trajectories starting from other angles (i.e. $<70^\circ$) make it to 80° . On the other end of the bending range, 5.8% of all trajectories reached less than 10° (i.e. 10° to 0°) which is still an order of magnitude more than that observed over the range from 70° to 80° . Therefore, our simulations encompassed extremities of bent conformations, even without starting from a low-probability model (i.e. 80°).

4. The influence of the hinge glycan N1242 on infectivity was analyzed using VSV-Luc reporter pseudovirus. While coronaviruses bud in ERGIC, where N-glycosylation gets further modified, VSV buds at the plasma membrane. In addition, coronaviral E protein has a function of intracellular retention, which helps concentrating viral structural proteins at the ERGIC to favor viral particle assembly. Therefore, the difference in budding and the missing E from the VSV pseudovirus can result in differences in glycosylation. The authors shall analyze the glycan compositions of the VSV pseudovirus, especially for the N1242 and N1247 sites, to control for the possible differences in glycosylation.

We concur with the reviewer's observation that the processing of glycoproteins incorporated in the pseudo-VSV may differ from those in coronaviruses due to the difference in virus budding sites. Our study employed the VSV pseudovirus system to investigate the role of N1242 and N1247 glycans in virus infection. Our findings revealed that the removal of the N1242 glycan specifically reduced pseudovirus infectivity, supporting our hypothesis that N1242 glycan plays a role in HCoV-NL63 infection based on structural and computational observations.

However, we respectfully disagree with the suggestion to conduct glycan analysis of the pseudo-VSV as we believe that such an approach would not provide additional evidence to support our conclusion. Our manuscript utilized an integrated approach using cryoEM, cryoET, MD, and

glycan analysis of the same HCoV-NL63 virion to arrive at our conclusions. We do not believe that pursuing a glycan analysis of the pseudo-VSV would yield new insights into coronavirus biology.

We are currently working on generating mutant HCoV-NL63 viruses using reverse genetics for an independent and rigorous investigation that goes beyond the scope of the current manuscript. Our aim is to conduct integrative biochemical and structural/computational analyses to further elucidate the role of N1242 and N1247 glycans in coronavirus infection. We believe that our current study is technically sound and biologically relevant to coronavirus researchers, given the global health concerns surrounding these viruses.

Minor points:

1. Figure 4D is never mentioned in the manuscript.

Thanks for the suggestion. Figure 4C and 4D together showed different epitope shielding by hinge glycans at different bending angles. We made the correction in the text.

2. Fig. S2B is not mentioned in the main text. There is no discussion on why/how two spikes are connected in the stalk.

Thanks for pointing this out and why/how two spikes are connected in the stalk is not clear but it is very rare. We removed Figure S2B in the revised manuscript.

3. Fig. 1G: the angles are boxed in colors, however, the colors are not explained.

Thanks for pointing this out and we removed colors of the angle boxes in the revised manuscript.

4. Adding a comparison animation of the MD simulation of S-trimers with and without the N1242 glycan to the manuscript will be highly appreciated.

We provided a supplemental movie (supplemental video 2) comparing the bending of HCoV-NL63 spike and of the spike with the two hinge glycans removed (Δ glycN1242/N1247) that has similar profile as spike with N1242 glycan removed (Δ glycN1242).

5. I'm wondering if the most favorable spike tilt 56° is the optimal tilt conformation, what would be the virological implications of this angle? Would the spike mediate receptor binding better so as to influence viral infectivity? A little discussion expansion on correlation between the spike tilting and viral infectivity will be appreciated.

We do not yet completely understand the mechanism of how the optimal spike tilt links to virus infectivity. The hypothesis of epitope accessibility and glycan shielding is purely structural. We are in the process of generating mutant HCoV-NL63 viruses for detailed structural, biochemical and virological analyses. In the discussion, we emphasized the necessity of such future studies.

Reviewer #2 (Remarks to the Author):

The manuscript by Chmielewski et al. provides important knowledge on the molecular dynamics of virion-bound HCoV-NL63 spike glycoprotein (S) and proposes that a specific N-glycan within a flexible hinge region at the stalk-crown interface confers some of the conformational flexibility. By applying focused classification, the authors were able to resolve the otherwise orientationally flexible stalk domain, indiscernible at the complete cryo-EM map. By further isolating image classes with various combinations of crown and stalk domain orientations, the authors also elegantly demonstrate highly variable tilting angles of virion-bound spike, with the crown domain not only tilting, but also rotating relative to the stalk domain, with this flexibility imposed by the hinge region. Intriguingly, in silico removal of the hinge-resident N-glycan at N1242, or all the S glycans, results in a shift of most energetically favorable tilting angles from 56° to around 25°, decreasing S flexibility, and possibly increasing exposure to antibodies. Moreover, HCoV-NL63 S-pseudotyped VSVdeltaG with said N-glycan deletion exhibits substantially decreased infectivity. The work provides extremely valuable insight into how glycans affect the conformational stability of proteins – a subject that is still underexplored. The authors provide compelling evidence for plausible roles of glycans in HCoV-NL63 biology, which could be further strengthened by in vitro experiments. The glycoproteomic analysis of virion-derived S is a welcome addition for determining predominant site-specific glycan compositions, however, a few details should be clarified.

1. The mass spectrometry sample and data analysis approach should be clarified. The multiple digests are typically employed to enhance coverage, and it is not unusual to pool them prior to analysis. Is that what was performed here, or were they analyzed separately? Given the semi-quantitative nature of spectral counting, were any measures taken to estimate variation (biological or technical replicates)?

The multiple digests were analyzed separately using the same instrument acquisition and data analysis method. After manual validation, the spectral counts from each digest were combined for quantitation. We consider the multiple digests as technical replicates.

2. The increased access to antibody molecules upon removal of N-glycans in the flexible hinge region makes sense, as suggested by the molecular modeling experiments (Figure 4C, 4D). However, the software-determined approach measuring the accessible area of the probe, and not of the exposed spike area is somewhat counterintuitive. It would strengthen the manuscript, if an

in vitro approach was employed to further test the validity of the claims. For example, suitable neutralizing antibodies, or sera from seropositive individuals could be used to measure antibody binding capacity, or neutralization of pseudotyped virions.

We'd love to do the suggested experiment if there is any neutralizing antibody targeting at the hinge loop available. Due to the lack of anti-HCoV-NL63 neutralizing antibodies, we can only predict the glycan mediated immune evasion using molecular dynamic simulation. We are planning to design molecular binders targeting the hinge loop and test their neutralizing activities against wt and glycan mutants. Like most structural results, they are used to guide better and functionally meaningful experiments.

3. Figure 5C. The plot elements should be described in the legend (measure of center, error bars, and statistical test). If the data is normalized to the infectivity of wild type, why are there error bars on the wild type data? What is the reproducibility of the data? Pseudovirus infectivity data can be rather variable, and the information on numbers of independent experiments is missing.

Thank you very much for pointing it out and we have revised the figure legend for Figure 5C accordingly. The bars represent means+SD of triplicate samples. The wild type infectivity was also tested in triplicate and the mean+SD of the wild type was shown. The mutants were normalized to the mean of the wild type. The experiments were repeated at least three times and data from one representative experiment was shown.

Minor:

Lines 254-255. The term “monosaccharide” should be used instead of “glycan”.

Thanks for the correction.

Line 319. It would be useful to mention exact numbers for glycan shield density for the other coronaviruses.

As suggested, we added panels G and H in Figure 3 showing the number of residues on the surface of different coronavirus spikes that are accessible to the Fab fragment of IgG. The less number of residues accessible to Fab, the more immune evasive the virus is. The analysis showed that HCoV-NL63 is more immune evasive (or glycan shielded) than other human coronaviruses.

Lines 484-485. How was the “conservation of glycosylation” determined as opposed to amino acid conservation (previous sentence)? Is it based on experimental data from the literature? If so, how many and which viruses? Or is it algorithm-based prediction?

We analyzed spike sequences from 35 different coronaviruses, at position 1242 and 1247, 89% and 74% sequences have ASN and at position 1244 and 1249, 91% and 69% of the sequences have either Ser or Thr. The conservation of N-glycosylation is based on the NxS/T sequence conservation at 1242-1244 and 1247-1249. We changed N-linked glycan to N-linked glycosylation sequence in the revised manuscript.

Figure S7D. Please add amino acid residue numbers for the start/end of the shown ranges.

Thanks for the suggestion, we added amino acid residue numbers in the sequence alignment shown in the new Figure S9D.

References Cited

- Ke, Zunlong, Joaquin Oton, Kun Qu, Mirko Cortese, Vojtech Zila, Lesley McKeane, Takanori Nakane, et al. 2020. "Structures and Distributions of SARS-CoV-2 Spike Proteins on Intact Virions." *Nature* 588 (7838): 498–502.
- Turoňová, Beata, Mateusz Sikora, Christoph Schürmann, Wim J. H. Hagen, Sonja Welsch, Florian E. C. Blanc, Sören von Bülow, et al. 2020. "In Situ Structural Analysis of SARS-CoV-2 Spike Reveals Flexibility Mediated by Three Hinges." *Science* 370 (6513): 203–8.
- Walls, Alexandra C., M. Alejandra Tortorici, Brandon Frenz, Joost Snijder, Wentao Li, Félix A. Rey, Frank DiMaio, Berend-Jan Bosch, and David Veasler. 2016. "Glycan Shield and Epitope Masking of a Coronavirus Spike Protein Observed by Cryo-Electron Microscopy." *Nature Structural & Molecular Biology* 23 (10): 899–905.
- Yao, Hangping, Yutong Song, Yong Chen, Nanping Wu, Jialu Xu, Chujie Sun, Jiaxing Zhang, et al. 2020. "Molecular Architecture of the SARS-CoV-2 Virus." *Cell* 183 (3): 730–38.e13.
- Zhang, Kaiming, Shanshan Li, Grigore Pintilie, David Chmielewski, Michael F. Schmid, Graham Simmons, Jing Jin, and Wah Chiu. 2020. "A 3.4-Å Cryo-Electron Microscopy Structure of the Human Coronavirus Spike Trimer Computationally Derived from Vitrified NL63 Virus Particles." *ORF Discovery* 1 (November): e11.

REVIEWER COMMENTS

Reviewer #1 (Remarks to the Author):

The authors have improved the manuscript, however, several concerns are still not fully addressed.

1. The authors have added the details for virus production and purification. They found a possible misinterpretation of the number of virions in their data. They re-checked the data, and based on a selected pool of 154 intact virions, their counts changed to 20 ± 13 spikes/virion from the original ~ 9 spikes/virion. This number is compared to that of SARS-CoV-2.

Despite the change of statistical analysis of the spikes/virion, the authors still report “On the virion surface, the average distance between the nearest neighboring spikes is ~ 34 nm (Figure 1E) compared to ~ 15 nm average nearest distance between prefusion spikes on SARS-CoV-2 virion” (line 175-178). Also, the purity of viral prep evidenced by the coomassie gel does not exclude the possibility that the spikes may detach from the virions during purification. Therefore, I am not fully convinced unless the authors control for the virion intactness (more specifically the No. of spikes/virion) by cryo-ET of freshly prepared & unconcentrated & not chemically fixed HCoV-NL63 in the supernatant.

2. The cryo-ET technical notes have now been improved. There are still confusions, e.g. supplementary table 2 says the Voltage is 200kV, however, the MM part says “Cryo-ET data was collected by loading frozen grids into a Thermo Fisher Titan Krios transmission electron microscope operated at 300kV”. Also, the tilt scheme shall be added to the table.

3. Why is the title of the manuscript “Integrated analyses reveal a hinge glycan regulates coronavirus spike tilting and virus infectivity” different from that of the SI “Bending of coronavirus spike regulated by a hinge glycan”?

Reviewer #2 (Remarks to the Author):

The authors have addressed my concerns.

Reviewer #3 (Remarks to the Author):

Chmielewski et al. presented a significant advance in understanding the structural dynamics of the spike protein of HCoV-NL63. Using cryo-ET and subtomogram averaging, the authors revealed a distribution of distinct titled conformations for the spike crown relative to its stalk region. As glycans are essential for the dynamics of spike, the authors used mass spectrometry to determine the site-specific occupancy and percentage of the N-linked glycan types at each site. The results revealed that different from other coronavirus such as SARS-CoV-2, the glycans on the spike of HCoV-NL63 are predominantly high-mannose glycans. The higher glycan shielding on HCoV-NL63 leads to less accessible residues for Fab fragment of antibody, suggesting that HCoV-NL63 is more immune evasive than other coronaviruses.

Using structural prediction tools like I-TASSER and AlphaFold, the author constructed a predicted model for the stalk region. MD simulations of the glycosylated spike at different tilt angles generated a similar profile of the spike bending dynamics to the results collected by cryo-ET. There are two glycans, N1242 and N1247, near the hinge region of the spike stalk. The authors found that these two glycans generated the most favorable protein-glycan interactions and the minimum accessible surface area around the most probable bending conformation (~56°) captured in the cryo-ET results. Removing these glycans, particularly N1242-linked glycan, shifted the distribution of bending conformations and decreased the protein-glycan contacts in the MD simulation. Finally, the authors found that mutation on N1242 (but not N1247) of spike reduced the infection of pseudo-VSV by ~70%.

This is a comprehensive study of the structural dynamics regarding the spike of HCoV-NL63 and provides important information on its glycans and immune evasion properties. However, there are some concerns that need to be addressed.

Major concern:

The evidence for structural dynamics is very compelling in the manuscript. However, the link between the structural dynamics of spike and infectivity is not convincing. Did the N1242D mutant spike generated on VSV-pseudovirus show different distributions on the bending conformations when compared to WT spike if measured by cryo-ET? Furthermore, the reduction of infectivity for N1242 mutant could be due to many reasons. Except for the MD simulation results (generated using the HCoV-NL63 spike, not the pseudo-VSV spike), there is no other evidence indicating the infection reduction is due to loss of the glycan-dependent motion. The response of the authors to Reviewer 1 brings more concerns – if ‘We do not believe that pursuing a glycan analysis of the pseudo-VSV would yield new insights into coronavirus biology’, then is the N1242D mutant spike generated using pseudo-VSV really a good model to validate the connection between the glycan-dependent motion and the infectivity? In the abstract, the authors quite emphasize the functional aspects of structural dynamics. For example, the abstract starts with ‘How the structure dynamics of the full-length spikes incorporated in viral lipid envelope correlates with the virus infectivity remains poorly understood.’ and ends with ‘Subsequent infectivity assays support the hypothesis that this glycan-dependent motion impacts virus entry.’ There is, in fact, no clear evidence linking the glycan-dependent motion to the virus entry in the manuscript. It is ideal that the authors can provide more evidence, but if the authors think it ‘goes beyond the scope of the current manuscript’, then at least they need to point out the limitation of the pseudo-VSV model

and tune down the description of the functional aspects (i.e., the infectivity) for the glycan-dependent structural dynamics.

Other concerns:

1) In the cryo-ET results, the authors found there are also different azimuthal directions along with different tilting angles. Because there is no preference for the azimuthal direction, the authors focused their analysis on the tilting angles. The authors did not state which azimuthal direction they used for the following MD simulations. Would different azimuthal directions impact the simulation result?

2) For the distributions of the bending angle probabilities (Fig. 4A and Supplementary Figure 10), the starting angle of 40° is quite close to the most probable angle observed in the cryo-ET results (~56°); however, the distribution peaks at around ~10° after MD simulation, any explanation for that?

Re: RESPONSES TO REVIEWER COMMENTS (MS # NCOMMS-23-05304A)

Reviewer #1 (Remarks to the Author):

The authors have improved the manuscript, however, several concerns are still not fully addressed.

1. The authors have added the details for virus production and purification. They found a possible misinterpretation of the number of virions in their data. They re-checked the data, and based on a selected pool of 154 intact virions, their counts changed to 20 ± 13 spikes/virion from the original ~ 9 spikes/virion. This number is compared to that of SARS-CoV-2.

Despite the change of statistical analysis of the spikes/virion, the authors still report “On the virion surface, the average distance between the nearest neighboring spikes is ~ 34 nm (Figure 1E) compared to ~ 15 nm average nearest distance between prefusion spikes on SARS-CoV-2 virion” (line 175-178). Also, the purity of viral prep evidenced by the Coomassie gel does not exclude the possibility that the spikes may detach from the virions during purification.

Therefore, I am not fully convinced unless the authors control for the virion intactness (more specifically the No. of spikes/virion) by cryo-ET of freshly prepared & unconcentrated & not chemically fixed HCoV-NL63 in the supernatant.

Responses:

The concern raised by the reviewer regards the potential fragility of coronavirus spikes leading to spike loss during virus purification, and consequently, the possible lack of representativeness in the observed spike dynamics on the virion surface. We respectfully disagree with this critique and offer the following rationales for our stance.

- A. The employed technique of gradient-purification followed by concentration of virions is a well-established biochemical protocol for preparing coronaviruses suitable for cryogenic electron microscopy (cryo-EM) and tomography (cryo-ET) investigations. This method has been successfully employed in the analysis of the structures of other coronaviruses, such as SARS-CoV, FCoV, MHV (Neuman et al., 2006), and PEDV (Huang et al., 2022). These previous studies, akin to our current approach, utilized virions purified through gradient ultracentrifugation, which effectively maintained the pre-fusion conformation of the spikes without experiencing S1 shedding during the purification process. It is important to note that this purification procedure is necessary

to yield enough virus particles suitable for high-resolution cryo-EM and cryo-ET structural analysis of spikes.

- B. We would like to underscore that HCoV-NL63 presents distinct challenges in terms of *in vitro* culture when juxtaposed with SARS-CoV-2, despite sharing the same receptor (Zhu et al., 2020). The diminished *in vitro* growth efficiency of HCoV-NL63 in comparison to other human coronaviruses that cause common cold, as noted by (Dijkman et al., 2013), makes it arduous to attain sufficiently high titer for direct imaging. It is impractical to carry out the cryo-ET experiment without concentration, as suggested by the reviewer.
- C. Importantly, the bending dynamics of HCoV-NL63 spikes predicted by the MD simulations are in good agreement with our cryo-ET observation of individual spikes on the purified virions in our study (Figure 4A). The quantitative agreement between experimental and *in silico* estimates of spike bending dynamics suggested our MD simulation system was aptly capturing the conformational ensemble of the spike *in situ*. This suggests that the bending dynamics of HCoV-NL63 spikes is intrinsic to the individual spike itself. We observed variations in spike numbers on the virion and distances between neighboring spikes. Thus, such biochemical variations observed in our experimental settings appear not to impact on the spike bending dynamics which is one of the primary conclusions in our paper.
- D. We thank the reviewer for accepting the corrected number of spikes/virion based on our first revision. However, with this correction, the spike-spike distance we reported will not change, as it is unrelated to the number of virions in the dataset that the reviewer correctly noted was incorrectly reported in the original manuscript Methods. Instead, the spike-spike distance analysis utilizes manually picked and refined spike subvolumes, where each virion in the tomograms was visually inspected in 2D slices for complete identification of all spikes prior to subtomogram processing. By utilizing refined subvolume orientations of spike stalks, which are anchored in the membrane, for determining nearest neighbor distances on the virion surface, our method represents an advanced quantification method for proteins attached to membranes. In contrast to earlier studies that approximated spike coordinates, our approach offers greater accuracy and is poised to contribute to future investigations into spike-antibody binding interactions and spike-receptor associations. Since there is no ground truth on the number of spikes and their spatial distributions known in any infectious coronavirus, our studies can only be validated by traditional biochemical assays like SDS gel and infectivity. Furthermore, the detailed infection steps are likely to be different between SARS-CoV-2 and HCoV-NL63, as exemplified by the lack of postfusion particles detected in our system as in SARS-CoV-2. HCoV-NL63 S protein is a single chain transmembrane glycoprotein with no cleavage site at S1/S2 boundary that is demonstrated by a single S protein band resolved by SDS-PAGE analysis of purified HCoV-NL63 (Fig. S3B in the

manuscript) in contrast to mixed S, S1 and S2 protein bands shown in SARS-CoV-2 (Ke et al., 2020). Trimerized HCoV-NL63 S proteins form spikes in single closed pre-fusion conformation that was observed in both purified soluble spike proteins (Walls et al., 2016) and *in situ* on purified virions (Zhang et al., 2020).

- E. Finally, we wish to clarify that we do not assert any claims regarding no potential loss of spikes from HCoV-NL63 virions during the standard purification protocol. While we acknowledge the reviewer's suggestion to examine unpurified CoV virions within CoV-infected cells, possibly involving various cell types and distinct biochemical conditions, as an important avenue of research, we concur that this lies beyond the confines of our current study's scope. Our primary message in this manuscript remains the detailed exploration of the structural dynamics of spikes on purified and concentrated HCoV-NL63 virions. Serendipitously, our analysis suggests that the bending of the spikes may modulate the infectivity of a pseudo virus. We believe that our study will inspire many follow-up experiments to validate our hypothetical model of the structure and function relationship of the spike *in situ*.

2. The cryo-ET technical notes have now been improved. There are still confusions, e.g. supplementary table 2 says the Voltage is 200kV, however, the MM part says “Cryo-ET data was collected by loading frozen grids into a Thermo Fisher Titan Krios transmission electron microscope operated at 300kV”. Also, the tilt scheme shall be added to the table.

We thank reviewer 1 for finding this inconsistency and have corrected the methods.

3. Why is the title of the manuscript “Integrated analyses reveal a hinge glycan regulates coronavirus spike tilting and virus infectivity” different from that of the SI “Bending of coronavirus spike regulated by a hinge glycan”?

We thank reviewer 1 for finding this inconsistency and have corrected the title of the SI. To tone down the claim, we changed our title to “Cryo-ET, Chemical and Computational Analyses Suggest That Hinge Glycans Modulate Coronavirus Spike Tilting and Infectivity”.

Reviewer #2 (Remarks to the Author):

The authors have addressed my concerns.

Response: We thank reviewer 2 for all the previous comments to improve the manuscript.

Reviewer #3 (Remarks to the Author):

Chmielewski et al. presented a significant advance in understanding the structural dynamics of the spike protein of HCoV-NL63. Using cryo-ET and subtomogram averaging, the authors revealed a distribution of distinct titled conformations for the spike crown relative to its stalk region. As glycans are essential for the dynamics of spike, the authors used mass spectrometry to determine the site-specific occupancy and percentage of the N-linked glycan types at each site. The results revealed that different from other coronavirus such as SARS-CoV-2, the glycans on the spike of HCoV-NL63 are predominantly high-mannose glycans. The higher glycan shielding on HCoV-NL63 leads to less accessible residues for Fab fragment of antibody, suggesting that HCoV-NL63 is more immune evasive than other coronaviruses.

Using structural prediction tools like I-TASSER and AlphaFold, the author constructed a predicted model for the stalk region. MD simulations of the glycosylated spike at different tilt angles generated a similar profile of the spike bending dynamics to the results collected by cryo-ET. There are two glycans, N1242 and N1247, near the hinge region of the spike stalk. The authors found that these two glycans generated the most favorable protein-glycan interactions and the minimum accessible surface area around the most probable bending conformation (~56°) captured in the cryo-ET results. Removing these glycans, particularly N1242-linked glycan, shifted the distribution of bending conformations and decreased the protein-glycan contacts in the MD simulation. Finally, the authors found that mutation on N1242 (but not N1247) of spike reduced the infection of pseudo-VSV by ~70%.

This is a comprehensive study of the structural dynamics regarding the spike of HCoV-NL63 and provides important information on its glycans and immune evasion properties. However, there are some concerns that need to be addressed.

Major concern:

The evidence for structural dynamics is very compelling in the manuscript.

Response: We thank the Reviewer for the overall positive evaluation of our work.

However, the link between the structural dynamics of spike and infectivity is not convincing. Did the N1242D mutant spike generated on VSV-pseudovirus show different distributions on the bending conformations when compared to WT spike if measured by cryo-ET? Furthermore, the reduction of infectivity for N1242 mutant could be due to many reasons. Except for the MD simulation results (generated using the HCoV-NL63 spike, not the pseudo-VSV spike), there is no other evidence indicating the infection reduction is due to loss of the glycan-dependent motion.

Response: We agree with the Reviewer that detailed analysis of a potential causal relationship between HCoV-NL63 infectivity and the bending of the spike is yet to be established. The correlation between these events is purely empirical as we have clearly stated in the

Limitations section about the modeling. Following the Reviewer's cogent thought, we have further toned down this narrative across the article. In addition, we changed our manuscript title to reflect our approach and conclusion as "Cryo-ET, Chemical and Computational Analyses Suggest That Hinge Glycans Modulate Coronavirus Spike Tilting and Infectivity".

The response of the authors to Reviewer 1 brings more concerns – if 'We do not believe that pursuing a glycan analysis of the pseudo-VSV would yield new insights into coronavirus biology', then is the N1242D mutant spike generated using pseudo-VSV really a good model to validate the connection between the glycan-dependent motion and the infectivity? In the abstract, the authors quite emphasize the functional aspects of structural dynamics. For example, the abstract starts with 'How the structure dynamics of the full-length spikes incorporated in viral lipid envelope correlates with the virus infectivity remains poorly understood.' and ends with 'Subsequent infectivity assays support the hypothesis that this glycan-dependent motion impacts virus entry.' There is, in fact, no clear evidence linking the glycan-dependent motion to the virus entry in the manuscript. It is ideal that the authors can provide more evidence, but if the authors think it 'goes beyond the scope of the current manuscript', then at least they need to point out the limitation of the pseudo-VSV model and tune down the description of the functional aspects (i.e., the infectivity) for the glycan-dependent structural dynamics.

Responses: Thank you for the question on the statement we used in the manuscript and the original rebuttal. We have modified the sentence in the revised abstract as the following: "Subsequent infectivity assays implicated involvement of N1242-glycan in virus entry". The assay we used is indeed a standard procedure to measure viral spike-mediated entry. Hope that our revision clarifies the misunderstanding due to our original wording.

Following the Reviewer's suggestions, we have further revised the article to ensure that no statements remain that structural dynamics of the spike are correlated with the infectivity assays.

Other concerns:

1) In the cryo-ET results, the authors found there are also different azimuthal directions along with different tilting angles. Because there is no preference for the azimuthal direction, the authors focused their analysis on the tilting angles. The authors did not state which azimuthal direction they used for the following MD simulations. Would different azimuthal directions impact the simulation result?

Responses: We thank the Reviewer for this insightful suggestion. Indeed, MD simulations were initially performed with a range of tilt angles but a single azimuthal angle which was arbitrarily

chosen. Our results were shown in Fig 4A in the manuscript. Subsequently, using 3.5 microsecond-long MD simulations, a broad range of azimuthal angles (between 0 to 60°) are sampled that are now presented in **Figure 1** below (Fig. S11 in the SI), which is also broader relative to the ones previously observed in MD simulations of SARS-CoV-2 spikes (Fig. 3B from (Kapoor et al., 2022)). However, given the higher diffusion barriers associated with azimuthal over tilting changes, the entire range of the azimuthal space (0 to 120° as seen in cryo-ET) is not captured in our finite time MD simulations. This limited sampling has further led to an artificial minimum around the azimuthal angle of ~20°, which will potentially even out with longer simulations. Nonetheless, as stated in the subsequent response the dynamics of the azimuthal angle is highly decoupled from that of the tilt. This decoupling between the two angular changes is observed from the maps, where despite an even distribution of the azimuthal angle, the tilt was prominently peaked around 56°. These caveats are now mentioned in the Limitations section.

The azimuthal angle controls the direction along which the bending occurs, while the tilt quantifies the magnitude of this bending. Our cryo-EM data shows that the spikes are dispersed so that the inter-spike dynamics (and hence the azimuthal changes) does not affect spike bending. In additional analysis that we now provide (**Figure 1**), we find that despite finite sampling there is little correlation between the azimuthal and tilt dynamics in the MD simulations.

Figure 1. Distribution of azimuthal and tilt angles sampled during MD simulations, and a scatter plot showing minimal correlation between the dynamics of these two angles during MD.

2) For the distributions of the bending angle probabilities (Fig. 4A and Supplementary Figure 10), the starting angle of 40° is quite close to the most probable angle observed in the cryo-ET results (~56°); however, the distribution peaks at around ~10° after MD simulation, any explanation for that?

Figure 2. Bending angle by replica for simulations using 40° map

Response: We have now investigated this data further. Presented in **Figure 2**, we find that two of the three replicas were indeed moving towards higher angles. There was one replica nonetheless that moved towards lower angles creating the peak. The cumulative area under the curve for drift towards 56° is still higher making the ~56° region more probable.

References Cited

- Dijkman, R., Jebbink, M. F., Koekkoek, S. M., Deijs, M., Jónsdóttir, H. R., Molenkamp, R., Ieven, M., Goossens, H., Thiel, V., & van der Hoek, L. (2013). Isolation and characterization of current human coronavirus strains in primary human epithelial cell cultures reveal differences in target cell tropism. *Journal of Virology*, *87*(11), 6081–6090.
- Huang, C.-Y., Draczkowski, P., Wang, Y.-S., Chang, C.-Y., Chien, Y.-C., Cheng, Y.-H., Wu, Y.-M., Wang, C.-H., Chang, Y.-C., Chang, Y.-C., Yang, T.-J., Tsai, Y.-X., Khoo, K.-H., Chang, H.-W., & Hsu, S.-T. D. (2022). In situ structure and dynamics of an alphacoronavirus spike protein by cryo-ET and cryo-EM. *Nature Communications*, *13*(1), 4877.
- Kapoor, K., Chen, T., & Tajkhorshid, E. (2022). Posttranslational modifications optimize the ability of SARS-CoV-2 spike for effective interaction with host cell receptors. *Proceedings of the National Academy of Sciences of the United States of America*, *119*(28), e2119761119.
- Ke, Z., Oton, J., Qu, K., Cortese, M., Zila, V., McKeane, L., Nakane, T., Zivanov, J., Neufeldt, C. J., Cerikan, B., Lu, J. M., Peukes, J., Xiong, X., Kräusslich, H.-G., Scheres, S. H. W., Bartenschlager, R., & Briggs, J. A. G. (2020). Structures and distributions of SARS-CoV-2 spike proteins on intact virions. *Nature*, *588*(7838), 498–502.
- Neuman, B. W., Adair, B. D., Yoshioka, C., Quispe, J. D., Orca, G., Kuhn, P., Milligan, R. A., Yeager, M., & Buchmeier, M. J. (2006). Supramolecular architecture of severe acute respiratory syndrome

- coronavirus revealed by electron cryomicroscopy. *Journal of Virology*, 80(16), 7918–7928.
- Walls, A. C., Tortorici, M. A., Frenz, B., Snijder, J., Li, W., Rey, F. A., DiMaio, F., Bosch, B.-J., & Veerler, D. (2016). Glycan shield and epitope masking of a coronavirus spike protein observed by cryo-electron microscopy. *Nature Structural & Molecular Biology*, 23(10), 899–905.
- Zhang, K., Li, S., Pintilie, G., Chmielewski, D., Schmid, M. F., Simmons, G., Jin, J., & Chiu, W. (2020). A 3.4-Å cryo-electron microscopy structure of the human coronavirus spike trimer computationally derived from vitrified NL63 virus particles. *QRB Discovery*, 1, e11.
- Zhu, N., Wang, W., Liu, Z., Liang, C., Wang, W., Ye, F., Huang, B., Zhao, L., Wang, H., Zhou, W., Deng, Y., Mao, L., Su, C., Qiang, G., Jiang, T., Zhao, J., Wu, G., Song, J., & Tan, W. (2020). Morphogenesis and cytopathic effect of SARS-CoV-2 infection in human airway epithelial cells. *Nature Communications*, 11(1), 3910.

REVIEWERS' COMMENTS

Reviewer #1 (Remarks to the Author):

Filtering the coronavirus through the membrane filter is unconventional and can detach the spikes. The control experiment to clarify this is cryo-ET of unconcentrated HCoV-NL63 in the supernatant, images of a few virions would do. According to our experiences, this can be done on SARS-CoV-2 and other seasonal human coronaviruses. If the authors can not supply this, they shall add a statement of the above risks to the Limitations part.

I have no further comments to the other parts.

Reviewer #3 (Remarks to the Author):

The authors addressed all my concerns in the revised manuscript.

Re: RESPONSES TO REVIEWER COMMENTS (MS # NCOMMS-23-05304B)

Reviewer #1 (Remarks to the Author):

Filtering the coronavirus through the membrane filter is unconventional and can detach the spikes. The control experiment to clarify this is cryo-ET of unconcentrated HCoV-NL63 in the supernatant, images of a few virions would do. According to our experiences, this can be done on SARS-CoV-2 and other seasonal human coronaviruses. If the authors can not supply this, they shall add a statement of the above risks to the Limitations part.

I have no further comments to the other parts.

Responses:

We want to thank this reviewer for the suggestion. Unfortunately, the virus titer in unconcentrated culture supernatant of HCoV-NL63 infected cells is too low to direct image without virus concentration/purification. We added this limitation in the discussion of the revised manuscript.

Reviewer #3 (Remarks to the Author):

The authors addressed all my concerns in the revised manuscript.

Responses:

Thanks.